# SCUBA: Salesforce Computer Use Benchmark

**Yutong Dai, Krithika Ramakrishnan, Jing Gu, Matthew Fernandez, Yanqi Luo,**
**Viraj Prabhu, Zhenyu Hu, Silvio Savarese, Caiming Xiong, Zeyuan Chen, Ran Xu**
Salesforce Research
{yutong.dai, kkrithika, zeyuan.chen, ran.xu}@salesforce.com

## ABSTRACT

We introduce SCUBA[1], a benchmark designed to evaluate computer-use agents on customer relationship management (CRM) workflows within the Salesforce platform. SCUBA contains 300 task instances derived from real user interviews, spanning three primary personas—platform administrators, sales representatives, and service agents. The tasks test a range of enterprise-critical abilities, including Enterprise Software UI navigation, data manipulation, workflow automation, information retrieval, and troubleshooting. To ensure realism, SCUBA operates in Salesforce sandbox environments with support for parallel execution and fine-grained evaluation metrics to capture milestone progress. We benchmark a diverse set of agents under both zero-shot and demonstration-augmented settings. We observed huge performance gaps in different agent design paradigms and gaps between the open-source model and the closed-source model. In the zero-shot setting, open-source model powered computer-use agents that have strong performance on related benchmarks like OSWorld only have less than 5% success rate on SCUBA, while methods built on closed-source models can still have up to 39% task success rate. In the demonstration-augmented settings, task success rates can be improved to 50% while simultaneously reducing time and costs by 13% and 16%, respectively. These findings highlight both the challenges of enterprise tasks automation and the promise of agentic solutions. By offering a realistic benchmark with interpretable evaluation, SCUBA aims to accelerate progress in building reliable computer-use agents for complex business software ecosystems. The code-base is

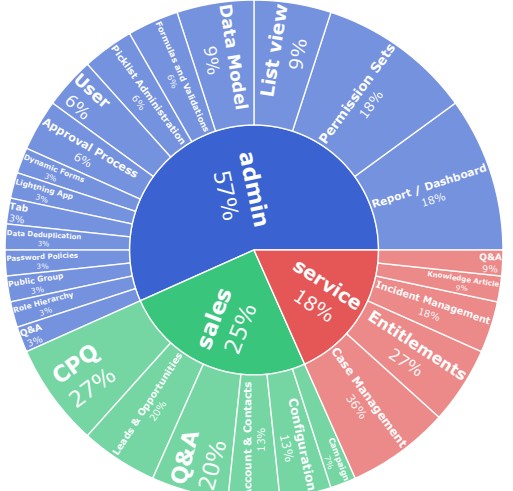

(a) Tasks distribution in different domains.

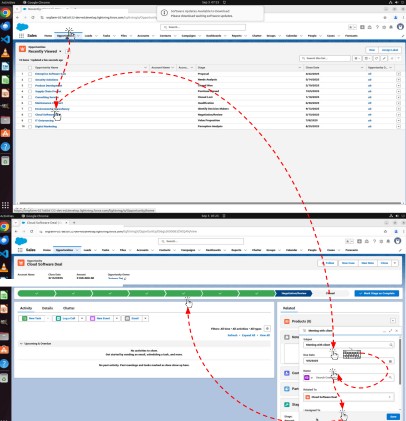

(b) One (simplified) agent trajectory on a task.

Figure 1: SCUBA tasks, environment, and agent trajectory preview.

---

[1]The code-base is available at https://github.com/SalesforceAIResearch/SCUBA.

# 1 INTRODUCTION

The advancement of large vision-language models (VLMs) has sparked increasing interest in turning these models into autonomous agents to automate different tasks, ranging from coding and deep (re)search to workflow automation (Yang et al., 2025a; Zhang et al., 2024b; Google, 2025; Claude, 2024; OpenAI, 2025c). One particular direction is to make these VLM-powered agents automate complex workflows via graphical user interfaces (GUIs) (Qin et al., 2025; Wang et al., 2025b; OpenAI, 2025c; Claude, 2024). Despite significant progress, the adoption of these agents in enterprise software environments remains a challenge and the performance of these agents is not fully understood. For performance, we mean more than success rates since in real business deployment, latency and costs are also decisive.

A primary obstacle is the lack of benchmarks that accurately reflect the multifaceted nature of enterprise software and workflows. Seminal benchmarks WebArena (Zhou et al., 2023) and OSWorld (Xie et al., 2024) are influential in evaluating agents on Web navigation and generic desktop application tasks, but they do not capture the complexities and challenges of using enterprise-grade platforms and tasks. WorkArena series (Drouin et al., 2024; Boisvert et al., 2024) are proposed to address this issue, with the environment and tasks built upon the ServiceNow platform. However, these tasks are often limited to service use cases. CRMArena series (Huang et al., 2025a;b) are built on the Salesforce platform, however, the tasks are mainly designed for tool use agents (Zhang et al., 2025a) and are based on information retrieval, i.e., the tasks do not write or change data records and settings in the environment. To bridge these gaps, we introduce SCUBA (Salesforce Computer Use BenchmArk), a comprehensive and realistic benchmark designed to rigorously evaluate computer-use agents on customer relationship management workflows. The comparison with related works is made in Table 1. Our key contributions are summarized below.

1. SCUBA is **realistic**. The tasks are derived from user interviews, covering the critical business functions of platform administration, sales, and customer service. These tasks aim to evaluate a spectrum of agents' abilities, including enterprise software UI navigation, data manipulation, workflow automation, information retrieval, and troubleshooting in the live Salesforce sandbox environments.

2. SCUBA provides multi-dimensional evaluation harness. Each task is paired with a rule-based evaluator, which not only tells whether the task is completed successfully, but also offers fine-grained **milestone score**, also known as the process reward (Zhang et al., 2025c). Based on the milestone scores, the evaluator also provides details on which part(s) of the task the agent failed to perform. SCUBA also provides metrics on the **latency** metrics and **costs** metrics.

3. SCUBA is shipped with **knowledge articles** and **human demonstration**, which can be utilized to improve agents' performance. We leverage human demonstration to simultaneously increase the task success rates and lower the latency and costs for a collection of agents.

4. SCUBA is efficient to use. We developed an **asynchronous evaluation pipeline**, with which the full evaluation can be finished in less than 90 minutes[2].

Table 1: Comparison on enterprise-task-oriented benchmarks. SCUBA uniquely combines all these aspects, while other benchmarks remain narrower in scope.

| Benchmark | # Instances (# Templates) | has human annotations? | Process Rewards? | Read&Write Tasks? | More than Service Tasks? |
|---|---|---|---|---|---|
| WorkArena | 19912 (33) | ✗ | ✗ | ✓ | ✗ |
| WorkArena++ | 682 (341) | ✗ | ✗ | ✓ | ✗ |
| CRMArena | 1170 (9) | ✗ | ✗ | ✗ | ✗ |
| CRMArena-Pro | 2140 (22) | ✗ | ✗ | ✗ | ✓ |
| SCUBA | 300 (60) | ✓ | ✓ | ✓ | ✓ |

# 2 RELATED WORK

## 2.1 COMPUTER-USE AGENTS

Due to the rapid growth of the relevant works on this topic, we do not attempt to conduct a comprehensive review and direct the interested readers to the survey (Zhang et al., 2024a) and the references

---

[2]This is for the self-served models. For API-based models, the bottleneck is on the API rate limits.

therein for more details. Instead, we focus on recent development of computer-use agents powered by vision-language models (VLMs), where the main inputs are the screenshot of the environment, and the outputs a sequence of executable actions, like clicking, typing, or even code blocks.

There are mainly three paradigms for developing computer-use agents of this kind. The first paradigm is to train a *foundational VLM with the computer-use as one of its capabilities*. Representative works include `Claude` family (Claude, 2025) and `QWen2.5-VL` (Bai et al., 2025). The computer-use ability is achieved by leveraging the VLMs' visual perception&grounding capability and the tool-use capability. Usually a computer-use toolbox is provided to regulate the model's action space, i.e., the output can only be valid executable actions. The second paradigm is to build a single native GUI agent, where the model is trained to have visual perception, visual grounding, reasoning, planning, action generation, and memory capabilities. `Aguvis` (Xu et al., 2025), `UI-TARS` family (Qin et al., 2025; Wang et al., 2025a), and `OpenCUA` (Wang et al., 2025b) belong to this category. The third paradigm is to build an agentic framework by offloading different capabilities such as visual grounding, reasoning, and planning to dedicated agents, which is in contrast to the second paradigm. Since the agentic framework admits special design, such as crafting special visual grounding agents (Agashe et al., 2024; 2025; Yang et al., 2025c; Xie et al., 2025) or enhancing GUI actions with coding and API calls (Song et al., 2025; Lai et al., 2025), this paradigm tends to outperform other two on benchmarks like `OSWorld` (Xie et al., 2024).

## 2.2 COMPUTER-USE BENCHMARKS

Challenging benchmarks are proposed to evaluate agents' capabilities from different dimensions, and the number continues to grow. We focus on benchmarks that are computer-use-centric and satisfy the following requirements: 1) the environments are interactive, where the agent is free to explore; 2) tasks can be tested with real **desktop operating systems** and/or **desktop browsers**; 3) reliable rule-based evaluators are available. These restrictions lead to the following classification.

**Browser Native Benchmarks.** `WebArena` and `VisualWebArena` are seminal benchmarks, where agents are required to navigate through simulated websites to perform tasks such as information retrieval, social media posting, and shopping. To account for the UI gap and modern website design in real-world websites, `REAL`(Garg et al., 2025) replicates the functionality and UI of widely used consumer platforms, for example, e-commerce, and professional networking. `WorkArena`-series (Drouin et al., 2024; Boisvert et al., 2024) and `Amazon-Bench` (Zhang et al., 2025b) adopts the real / sandbox of production-grade websites to build tasks. `WorkArena` focus on the service-related tasks on the ServiceNow platform while `Amazon-Bench` puts emphasis on testing agents' capability on various functionalities offered by the Amazon shopping website.

**OS Desktop Native Benchmarks.** `OSWorld` (Xie et al., 2024) is the first real computer environment, using 9 desktop applications to cover a variety of computer tasks, such as manipulation of slides, documents, images, and excel tables. It mainly focuses on the Ubuntu OS. Follow-up works `WindowsAgentArena` (Bonatti et al., 2024), `macOSWorld` (Yang et al., 2025b), and `MMBench-GUI` (Wang et al., 2025c) extend the efforts to cover tasks on Windows and MacOS and expand the evaluation dimensions to include the safety and efficiency.

## 3 SCUBA BENCHMARK

`SCUBA` is an interactive environment built on the Salesforce platform, a widely used cloud-based enterprise platform for CRM workflows. `SCUBA` is designed to cover three major types of tasks that are highly valuable for automation. Each task is paired with an initialization script and rule-based evaluation methods. Details are discussed in the following sub-sections.

### 3.1 ENVIRONMENT

**Realism** is one of the key requirements when designing `SCUBA`, therefore, we opt to use a sandbox developer org offered by Salesforce[3]. Its UI is identical to production orgs and free-tier features are sufficient to support all tasks. However, the pursuit of realism also brings unique challenges for

---

[3]The public can sign up for a sandbox org via the link `https://www.salesforce.com/form/developer-signup/?d=pb`

resetting the environment. For benchmarks like `WebArena` (Zhou et al., 2023) and `OSWorld` Xie et al. (2024), resetting can be done by restarting Docker containers. For a Salesforce sandbox developer org, resetting the org to its initial state is equivalent to creating a new one, which is prohibitive for supporting large-scale evaluation. Therefore, we address this via taking a snapshot of the initial states of the org by downloading a set of configuration files. During the resetting phase, we compare the differences of the initial states of org and modified states after an agent run. We then revert the configuration files that are changed by the agent to their initial states. This means that the reset is done on the task level since we know what configurations are expected to be changed by agents.

**Parallelism** is another key requirement to allow efficient agent interaction and evaluation. Therefore, we build infrastructures to support this. Users can choose to spin up either a desired number of browser sessions or OS desktops (running in the Docker containers) concurrently. So, a large collection of agents can be assigned to dedicated environments and can be tested asynchronously on `SCUBA` to save the time.

**Observation & Action Space.** The environment supports screenshots, accessibility trees, and flatten DOM object strings as the primary observation for the agent. All primitive actions supported by Playwright[4] and PyAutoGUI[5] can be used to interact with the environment.

## 3.2 TASKS

This version of `SCUBA` contains 300 test cases, covering daily workflows from three types of personas, namely, the platform administrator, the sales role, and the service role. The tasks are sourced by interviewing real Salesforce platform users, including the solution engineers, org administrators, sales development representatives, business development representatives, and account executives. After collecting the use cases, we prepare the tasks by associating each with one knowledge article on Salesforce Trailhead[6], a learning platform that provides tutorials on how to use Salesforce platform. The distribution of tasksis visualized in Figure 1(a).

### 3.2.1 CORE ABILITIES

The tasks are designed to test the following abilities of the agents.

**1. Enterprise Software UI understanding.** Navigating on enterprise software to find correct pages to complete the task can already be very challenging for a daily computer user. The UI design is typically more minimalistic and prioritizes robustness and efficiency. Usually, formal training with detailed on-boarding materials is required to help users become familiar with the Salesforce platform. Therefore, agents need to have strong enterprise software UI understanding, navigation, and interaction abilities. For example, the task below requires a human agent to navigate to 8 different pages with 31 steps to complete. *"Create a territory model with the name "by Finance Industry" and with the rule name "Industry, Active, and Type". The selection criteria are "Industry equals Finance", "Account equals Active", and "Type is not equal to Channel/Partner Reseller or Installation Partner". If the sales territory is not enabled, enable it first."*

**2. Information Retrieval & Textual Reasoning.** Service and Sales users usually want to quickly look up some quick facts buried in thousands of records. We build tasks to test agent's ability to achieve this goal. One sample task looks like the following. *"I'm reviewing the call transcripts to extract useful information. Please examine the call between [Person A] and [Person B], and tell me within how many days does the [Person A] wish to have the installation done. Respond with the number only."* In this query, [Person A] and [Person B] are made up user records, but we choose to mask out to avoid violation of the double-blind policy.

**3. Salesforce Data Manipulation.** One of the core functionality of the Salesforce platform is about managing data securely. Tasks on the data model, metadata, and data records creation/update/deletion need to be handled with correct privilege. The Salesforce platform provides a set of features to ensure the standards obligation. We design tasks like the following to test agents' such ability. *"Add a formula field with the name "Days to close" to the object Opportunity. It tracks the number of days until an Opportunity Closes. This newly added field should only be visible to Marketing User."*

---

[4] https://playwright.dev/python/
[5] https://pyautogui.readthedocs.io/en/latest/
[6] https://trailhead.salesforce.com/

**4. Business Workflow Build & Execution.** One critical usage of Salesforce platform is to build and automate some established business workflows, where things must be done in a prescribed order following a certain policy. One sample task is "*Configure the Approve Opportunity Amount process's final approval actions by adding a new action with the name 'approve when successful', which sets the amount approval status field value to 'Approved'. Similarly, add a new action under the final rejection actions with the name 'reject when failed', which sets the field value to 'Rejected'. If the amount approval status field does not exist in the Opportunity object, create it first.*"

**5. Q&A for troubleshooting.** Our survey reveals that platform administrators and human service agents need to ground on the org's data to answer client questions. Therefore, we design tasks to test if the agents can answer questions faithfully. A sample task test this ability is as follows. "*A user is assigned with Account Access permission set. Will the user be able to delete any records under the Account Object? Based on the Salesforce org's object settings, only answer with "yes" or "no".*"

### 3.2.2 TASK CONSTRUCTION DETAILS

In this sub-section, we describe the pipeline for the task construction.

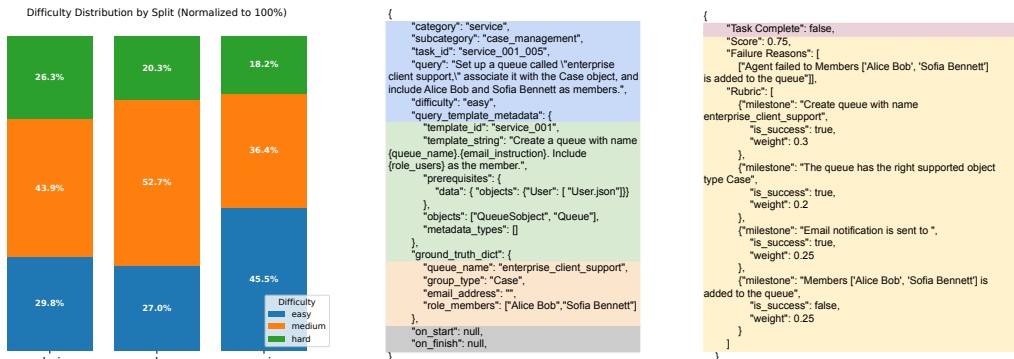

Figure 2: **Left**: Difficulty distribution by splits. **Middle**: A sample task configuration files. The light blue section contains basic information on the task; the light green section docuements details for initialization; the light orange section highlights the inputs used for rule-based evaluation; the gray section can be used to manipulate the environment before and after agent run, leaving the freedom to configure different initial states and perform post-process. **Right**: A sample evaluation result.The light purple section indicates if the task is successful. The light yellow section lists detailed milestone scores and rubrics.

**Phase 1: Task Template Creation.** For each workflow obtained from the user research, we build a template for it, e.g., "*Configure the organization-wide defaults for the {object_name} object with visibility {internal_visibility} for internal users, visibility {external_visibility} for external users, and allow users higher in the role hierarchy.*" [7] Each template is paired with one knowledge article[8] from the Trailhead. A complexity level from the set {easy, medium, hard} is assigned to the task template based on UI actions complexity and unique abilities required to finish the task. The distribution of difficulty levels by splits is visualized in Figure 2.

**Phase 2: Task Queries Creation.** For each task template, we generate five distinct task queries by filling in the template with different values. These values are deliberately selected so that queries from the same template vary in difficulty. For instance, one query might require extra scrolling or alphabet-based searching to locate the target data, while another might demand keeping track of more previously deselected check boxes. Then, the 5 instances are paraphrased by GPT-5 to mimic the real human queries and increase the language diversity. Finally, two authors manually check all paraphrased queries to ensure language diversity, wording accuracy, and ethics compliance. If any paraphrased query failed to meet the standards, the authors would rewrite it before adding it to the benchmark.

---

[7]The values within the curly brackets can be replaced with the valid values.

[8]For the example used here, the link is https://trailhead.salesforce.com/content/learn/modules/lex_implementation_user_setup_mgmt/lex_implementation_user_setup_mgmt_configure_user_access-hoc

**Phase 3: Task Initialization & Evaluation.** Most of the tasks in SCUBA require prerequisites. These prerequisites include, but are not limited to, synthesizing and uploading data records that the tasks depend on and enabling the org's settings such that certain permissions are granted to allow tasks to be completed. An example can be found in the middle subplot of Figure 2. We also manually craft evaluation methods for each task query using Salesforce's APIs, for example, Metadata APIs[9] and Tooling APIs[10]. The evaluator not only offers a binary 0/1 success score, but also provides milestone scores (process reward) with rubrics. This provides insights into the particular parts where the agent failed during execution. An example can be found in the right subplot of Figure 2.

**Phase 4: Human Annotation** In the final phase, all instances are sent to the internal annotation team to annotate. Each annotator is provided with an internal annotation tool, a review tool, and a set of task instances. The annotation process serves several purposes. 1) *Quality Control.* This process ensures that all task instances are valid so that they can be completed by human. In addition, this process helps to test whether the results of the evaluators match with the human expectation. 2) *Enhance the experiments.* Annotated trajectories are used to to potentially improve agents' performances (discussed in Section 4). More details on annotation process can be found in Appendix F.

## 4 EXPERIMENTS

### 4.1 SETUPS

Since both browser-use agents and computer-use agents can be used to finish the tasks in SCUBA. Here we describe the setups.

Table 2: Comparisons of the setups for two types of agents. A total of 9 agents are tested.

| Agent Type | Observation Space | Action Space | Backbone Models |
|---|---|---|---|
| Browser-Use | SOM + DOM texts | Table 4 | GPT-5, o3, Claude-4-sonnet, Gemini-2.5-pro |
| Computer-Use | Screenshot | Table 5 | UI-TARS-1.5, OpenCUA, Agent-S2.5, OpenAI-CUA, Claude-4-sonnet |

**Formalization.** Each task of SCUBA can be formulated as a partially observable Markov decision process $(\mathcal{S}, \mathcal{A}, \mathcal{T}, \mathcal{R}, \Omega, \mathcal{O})$. $\mathcal{S}$ is the environment states space and $\mathcal{A}$ is the agent's action space. $\mathcal{T}(s'|s, a) : \mathcal{S} \times \mathcal{S} \times \mathcal{A} \rightarrow [0, 1]$ is the state transition probability function that describes the conditional probability of changing to state $s' \in \mathcal{S}$ given the action $a \in \mathcal{A}$ and the state $s \in \mathcal{S}$. $\mathcal{R}(s, a) : \mathcal{S} \times \mathcal{A} \rightarrow [0, 1]$ is a reward function. $\mathcal{O}$ is the observation space and $\Omega(o' \mid s', a) : \mathcal{O} \times \mathcal{S} \times \mathcal{A} \rightarrow [0, 1]$ is a probability function describing the conditional probability of observing $o' \in \mathcal{O}$ after taking the action $a \in \mathcal{S}$ and moving to the states $s' \in \mathcal{S}$. Denote the agent (a generative model) as $\pi$ and the task query as $Q$. At the time stamp $t$, the task for the agent is to sample an action as $a_{t+1} \sim \pi(a \mid H_t, Q)$ with $H_t = \{o_0, a_1, o_1, \dots, a_t, o_t\}$. The goal is to maximize the rewards.

**Observation ($\mathcal{O}$) & Action ($\mathcal{A}$) Space.** For browser-use agents, we choose the Set-of-Mark (Yang et al., 2023) (SOM) and DOM texts as the observation $o$ to approximate the environment state $s$. SOM places bounding boxes with numeric tags on the interactive elements on the screenshot of the browser's viewport. Each DOM text contains the index, attribute, and label of an element. The observation $o$ for computer-use agents is just the screenshots of the entire OS desktop. Please refer to Appendix B for examples. The action space for browser-use (19 actions) and computer-use agents (15 actions) are described in the Appendix C. We remark that, since different computer-use agents may have different action space by design, we eventually map them to a unified space in Table 5.

**Backbone models ($\pi$).** Under the constraints of access to different API services and the computation resources to serving models, we make the following decisions about the backbone models to test. For browser-use agents, we adopt GPT-5 (OpenAI, 2025a), o3 (OpenAI, 2025b), Claude-4-sonnet (Claude, 2025), and Gemini-2.5-pro (Comanici et al., 2025). For computer-use agents, since there are many candidates, we choose performant ones according to the OSWorld leader board. Under the foundational VLM category, we

---

[9]https://developer.salesforce.com/docs/atlas.en-us.api_meta.meta/api_meta/meta_intro.htm
[10]https://developer.salesforce.com/docs/atlas.en-us.api_tooling.meta/api_tooling/reference_objects_list.htm

choose `Claude-4-sonnet` (Claude, 2024) with the computer toolbox. Under the native GUI agent, we choose `UI-TARS-1.5-7B` (Qin et al., 2025), `OpenCUA-7B` (Wang et al., 2025b), `OpenAI-CUA` (OpenAI, 2025c), and `GUI-Owl-7B` (Ye et al., 2025). Under the agentic framework category, we choose `Agent-S2.5` (Agashe et al., 2025) and `MobileAgentV3` (Ye et al., 2025) (paired with `GUI-Owl-7B`).

Finally, we summarize the above discussion in Table 2 for easy comparison.

## 4.2 IMPLEMENTATION DETAILS

Table 3: Performances of various agents on the `SCUBA`. All metrics are averaged accross all instances. ↑ next to each column name represents the larger value, the better and vice-versa for ↓.

| Method Name | Milestone Score ↑ | Success Rate ↑ | Time (min) ↓ | Steps ↓ | Tokens (k) ↓ | Costs ($) ↓ |
|---|---|---|---|---|---|---|
| **Zero-shot Setting** | | | | | | |
| Browser-Use Agents | | | | | | |
| GPT-5 | 0.73 | 51.33% | 19.31 | 22.9 | 245.24 | 0.55 |
| o3 | 0.65 | 45.67% | 21.37 | 26.74 | 237.16 | 0.56 |
| Gemini-2.5-Pro | 0.47 | 31.00% | 7.27 | 15.61 | 134.94 | 0.24 |
| Claude-4-sonnet | 0.56 | 34.67% | 11.25 | 24.12 | 353.47 | 1.16 |
| Computer-Use Agents | | | | | | |
| UI-TARS-1.5-7B | 0.10 | 2.67% | 6.14 | 35.43 | 183.69 | -[*] |
| OpenCUA-7B | 0.05 | 0.67% | 20.48 | 31.21 | 133.88 | -[*] |
| GUI-Owl-7B | 0.05 | 1.00% | 12.96 | 34.05 | 167.64 | -[*] |
| OpenAI-CUA | 0.29 | 16.00% | 5.80 | 31.32 | 193.83 | 0.59 |
| Claude-4-sonnet(computer) | 0.48 | 27.00% | 8.02 | 35.24 | 457.77 | 1.44 |
| MobileAgentV3 | 0.11 | 3.10% | 21.47 | 41.96 | 755.63 | 1.17[*] |
| Agent-S2.5(w/GPT-5) | 0.58 | 39.00% | 25.13 | 34.23 | 597.92 | 1.17[**] |
| **Demonstration-Augmented Setting** | | | | | | |
| Browser-Use Agents | | | | | | |
| GPT-5 | 0.75 | 53.85% | 17.76 | 21.33 | 232.76 | 0.51 |
| o3 | 0.69 | 50.00% | 17.05 | 25.33 | 236.63 | 0.55 |
| Gemini-2.5-Pro | 0.71 | 46.15% | 10.02 | 19.48 | 177.13 | 0.31 |
| Claude-4-sonnet | 0.68 | 46.15% | 11.21 | 23.92 | 364.10 | 1.19 |
| Computer-Use Agents | | | | | | |
| UI-TARS-1.5-7B | 0.24 | 9.16% | 6.52 | 37.65 | 213.07 | -[*] |
| OpenCUA-7B | 0.12 | 5.38% | 18.86 | 33.57 | 159.11 | -[*] |
| GUI-Owl-7B | 0.12 | 2.70% | 11.18 | 36.81 | 181.50 | -[*] |
| OpenAI-CUA | 0.50 | 28.85% | 5.29 | 27.29 | 171.93 | 0.52 |
| Claude-4-sonnet(computer) | 0.67 | 47.69% | 7.33 | 32.05 | 424.90 | 1.34 |
| MobileAgentV3 | 0.18 | 7.30% | 19.96 | 42.98 | 830.85 | 1.17[*] |
| Agent-S2.5(w/GPT-5) | 0.57 | 40.38% | 24.63 | 34.43 | 626.26 | 1.19[**] |

[*] We do not report the inference prices for the self-hosted models.

[**] This does not include the costs of serving `UI-TARS-1.5-7B` as the grounder used in `Agent-S2.5`.

**Agent Implementation.** We primarily focus on testing two types of agents, Browser-Use agents and Computer-Use agents. The implementation of the agent loop is adapted from the implementation proposed in `browser-use` (Müller & Žunič, 2024) and `OSWorld` (Xie et al., 2024). We made significant changes in the DOM parser provided by the `browser-use` to improve the quality of detecting all interactive elements on the Salesforce platform and handling iframe and cross-origin contents, which results in better quality of SOM screenshots and DOM texts. Without these changes, agents run with `browser-use` would perform significantly worse compared to the number we reported in Table 3. We also modify `OSWorld`'s agent implementation to better handle tasks from `SCUBA`, for example, system prompts.

**Asynchronous Parallel Inference Infrastructure.** `SCUBA` supports asynchronous parallel evaluation to reduce the overall evaluation time. The asynchronous support for browser-use agents is straightforward by spinning up a desired number of browsers in the headless mode and assigning each agent to a dedicated browser with one task. For computer-use agents, the asynchronous inference pipelines starts by creating a a fixed number of Docker containers on a remote server with each

container running a Ubuntu system. Whenever the docker container is not occupied by an agent, we immediately assign an agent with a task to it. The container first reverts the system to the initial snapshot, then the agent runs with a fresh desktop and a new task. Since there are more tasks than Docker containers, this allows the full utilization of resources.

**Parameters.** The maximum allowed steps, i.e., the number of predictions made by the agent is set to 50. This is justified by the average steps used from the human annotation, and we allow extra step buffers. The maximum execution time allowed per task is 90 minutes. We terminate the agent execution loop, when either 1) agent thinks the task is done, 2) the maximum steps budget is reached, or 3) the maximum time budget is reached. The resolution of the screenshot is set to $1920 \times 1080$ by following `OSWorld`. The generation temperature is set to $1.0$ since some models can only support the temperature set to $1.0$, e.g., `GPT-5`. We also empirically found that setting the temperature to $1.0$ prevents the agents from getting stuck on some particular pages. The agent observation $H_t$ is the concatenation of the actions performed up to the time stamp $t$ plus the current screenshot.

For extended details on the implementation, and metric calculation, please refer to Appendix G.

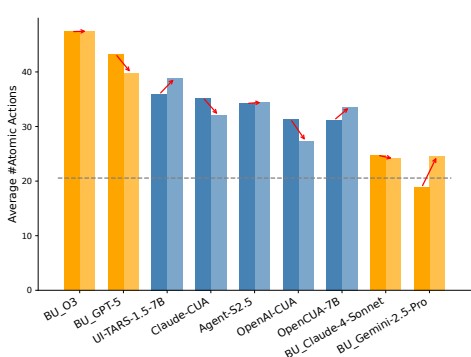

Figure 3: Average number of atomic actions used by different methods. Within each method, the left corresponds to the zero-shot setting and the right is the demonstration-augmented setting. The gray dashed lines is the human average.

### 4.3 MAIN RESULTS

Agents are tested in two settings, 1) zero-shot, where only the task query is used, and 2) demonstration-augmented, where the task query is combined with human demonstration for the similar or the same task. Considering the second setting is mainly motivated by the observation that agents may lack necessary domain knowledge about the UI design and functionalities of the Salesforce platform. Like using tutorials to train a real person to use the Salesforce platform, we prepare the demonstration for agents[11]. The results for both settings[12] are summarized in Table 3. Based on the experiment results, we present some observations and analysis.

**Demonstration is a effective way to improve performance metrics.** As shown in Table 3. The human demonstration can be an effective strategy to increase the agents' performance metrics across the boards. It can consistently improve task success rates, while lowering time consumption, number of steps, token consumptions, and costs. However, there are few exceptions. For computer-use agents `UI-TARS-1.5-7-B` and `OpenCUA-7B`, we see that while task success rates are improved, some other metrics also increase. We attribute these to the fact that these two agents cannot use the human demonstration effectively. Also, we empirically found that `UI-TARS-1.5-7-B` tends to provide answers for Q&A questions in Chinese characters, which affects its performance. For browser-use agents, there is one exception with `Gemini-2.5-Pro`. We attribute the increases in the number of steps and costs to the fact that the human demonstration has room to be improved since we observe that this agent can exploit some shortcuts that are absent from the human demonstration. It is worth mentioning that the number of steps may not capture the nuances that some agents can generate multiple atomic actions with one prediction. We further add a metric called # atomic actions (defined in the action space) and visualize it in Figure 3. Lastly, we can see from Table 3 and Table 6 (in the appendix), agents have varying abilities to leverage the human demonstration, and if the agents' tasks success rates are already very high, e.g, `GPT-5`, then the task success rates improvement may not be that significant.

**Design patterns: What type of agent should I use?** For enterprise use cases, while having high success rates is crucial, costs and latency are also important factors. To answer the question, we visualize the agents' performance metrics in Figure 4. Methods falling into the green zone are good

---

[11]Please refer to Appendix D for more details.

[12]We've conducted additional cost-constrained setting. The details are in Appendix I.

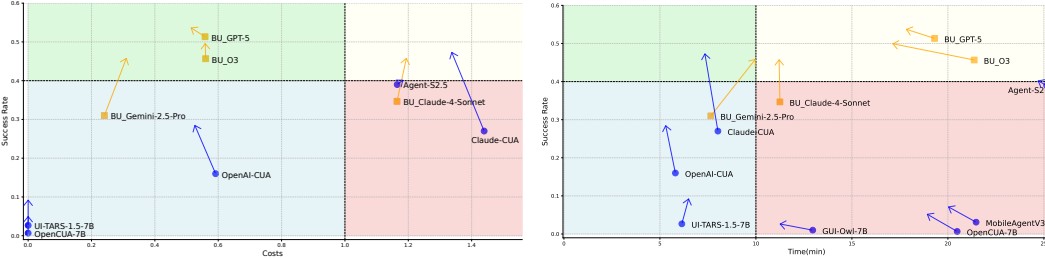

Figure 4: Left: Costs v.s. Success Rate. Right: Time v.s. Success Rate. Orange squares and blue circles represent the browser-use agents and computer-use agents' performance metrics under zero-shot setting. The arrow points the performance metrics under the demonstration-augmented setting. The arrow that points to top-left are desired, since it means improvements. Green zone means low costs/latency and high success rates zone; vice versa for the red zone. Some methods are omitted in the left plot as they overlapped (costs are zero).

candidates, since they have relatively high task success rates and low latency / costs. In general, we observe that *computer-use agents have lower latency than browser-use agents at the cost of lower success rates*. The high latency of browser-use agents comes from two parts, the API service response time and the multi-agent agent framework design[13]. If latency is not a concern, browser-use agents can be a better solution. The high task success rates seen in browser-use agents can be attributed to the combination of the stronger planning ability of the foundation models, richer observation space, and the more efficient action space design used in browser-use agents. Overall, we find that the browser-use agent powered by `Gemini-2.5-pro` (with demonstration) strikes the balance among the task success rates, latency, and costs.

**Directions to improve the computer-use agents.** Although browser-use agents have higher tasks success rates, the success is built upon the tons of meticulous customization of DOM parser for the Salesforce platform. The original DOM parser shipped with `browser-use` framework failed to capture some types of elements, making many tasks infeasible.

On the other hand, computer-use agents only require screenshots, which is easier to acquire robustly. Also, if comparing the performance of `Claude-4-sonnet` under both computer-use and browser-use settings, the performances are close, hinting that the computer-use agents still have their places. To further improve the computer-use agents, we should improve their generalization ability as shown in Figure 5. The performance drop is significant when moving from `OSWorld` to `SCUBA`. When analyzing the trajectories of computer-use agents, we found that planning and grounding are still the main causes of task failure. Planning can be partially addressed by providing the demonstration, and

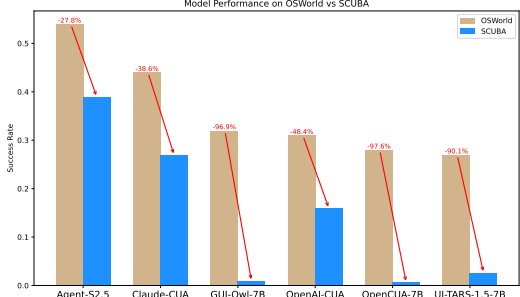

Figure 5: Performance drop when moving from `OSWorld` (50 steps) to `SCUBA` (without demonstration).

it is the primary source of the task success rate improvement. On the other hand, the grounding issue still persists, i.e., the agent failed to correctly predict the coordinates of elements with which it tends to interact. However, if the agents can make multiple predictions on the coordinates and use a majority-voting approach (Yang et al., 2025c), the grounding accuracy might also be improved. More qualitative analysis and discussion on solutions with examples can be found in Appendix H.

## 5 DISCUSSION

While we strive to provide a comprehensive evaluation on the computer-use agents' performance on the customer relationship management (CRM) workflows, with a special focus on the Salesforce Platform. This benchmark still has several limitations worth discussing.

---

[13]The performance for `Agent-S2.5` with `GPT-5` also validates this claim.

- **The rule-based evaluators only check whether the tasks are completed, yet fail to capture whether the tasks are completed in a trustworthy and safe manner.** To be precise, the evaluators in `SCUBA` are similar to `OSWorld`, which only check whether the environment's final state matches the prescribed states, yet can not detect whether the agent sneakily modified or deleted task-irrelevant data. Ideally, the robust approach to detecting the agent's unexpected or unsafe behavior is to compare the environment states before and after the agent runs. However, this approach is prohibitive since the environment state files are huge for the Salesforce Platform, and we can not download all of them, even depleting the daily API quota. So, comparing the before-and-after environments' state approach is not feasible. A compromised solution is to understand the failure modes of the agents and add case-by-case detection. We have the annotators inspect trajectories generated by a subset of agents and instructed the annotators to inspect whether the agents made any unintended changes to the environment, like adding data not asked for or accidentally deleting irrelevant data. We indeed observed that the agent may accidentally alter some relevant data fields that the task does not ask for. To address it, we add task-specific checks to capture this behavior. Admittedly, this is neither scalable nor comprehensive.

- **`SCUBA` only a subset of enterprise workflows.** Enterprise workflow involves many more scenarios than the tasks in this benchmark. According to our survey, a good portion of workflows involve multiple platforms. For example, a sales agent might use HubSpot for lead generation, and then use LinkedIn Sales Navigator for decision role confirmation, and finally manage lead information in the Salesforce platform. Even just within the Salesforce platform, there are additional tasks from the survey not included in `SCUBA`, for example, lead scoring and AI-driven insights. These tasks are not included in `SCUBA` because either the functionalities used in the workflows are paid features or there are no freely available sandbox environments (and the subscription fee for some platforms can be significant).

## 6 CONCLUSION

In this work, we introduced `SCUBA` to evaluate computer-use agents on realistic CRM tasks within the Salesforce platform. `SCUBA`'s primary contribution is its focus on realistic enterprise-grade workflows across diverse business roles, addressing a critical gap left by existing benchmarks. Our experiments revealed two significant discoveries. First, browser-use agents' observation and action space design can better leverage the foundation VLMs' capabilities and outperform specialized computer-use agents at the current stage. Also, a substantial performance gap exists between current open-source and closed-source models. Second, we showed that augmenting tasks with demonstrations is a highly effective strategy, improving success rates while simultaneously reducing task time and cost. These findings highlight both the current limitations and the promising future directions for developing capable autonomous agents for complex business software, especially on *how to leverage the unstructured documents and tutorials to more effectively since structured human demonstration is not always available*.

**Acknowledgment**  We thank Chien-Sheng (Jason) Wu and Kung-Hsiang (Steeve) Huang for sharing their insights on developing the `CRMArena` benchmark series (Huang et al., 2025a;b). We are also grateful to Nesrine Yakoubi, Fabriana Louisa Pita, Michael Thuo, and Caitlyn Cline for their annotation support.

**Ethics statement**  The authors confirm that we have carefully reviewed and adhered to the ICLR Code of Ethics in this work.

**Reproducibility statement**  Our codebase is accessible via this link[14] to ensure reproducibility. However, audience should be aware that the Salesforce developer org sandbox is an evolving environment and is maintained by Salesforce. Therefore, there might be breaking changes we cannot control and might make some evaluators or tasks problematic, thus hurting reproducibility. The authors cannot address this by fixing a certain version of the sandbox. Also, the authors have no control over the closed-source model's API service. However, the authors will strive for the best efforts to make the results reproducible and promise to actively maintain the benchmark. If some tasks become invalid due to the environment change, we will replace them with similar tasks.

---

[14]https://github.com/SalesforceAIResearch/SCUBA

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

## A    LLM USAGE STATEMENT

The large language model is used for the manual script proofreading to detect grammatical errors and rephrase some overly long sentences. As described in the paper, the LLM (GPT-5) is used to paraphrase the task queries and summarize human annotations. VLMs (see a list of them in Table 2) are used in the experiments.

## B    AGENT OBSERVATION SPACE

We provide an example of SOM + DOM texts used as the observation space for the agents running with `browser-use`.

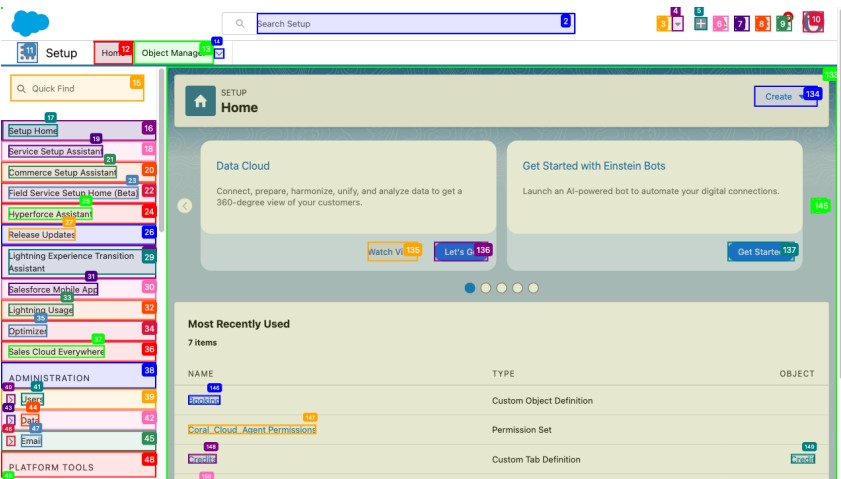

(a) Set-of-Marks Screenshot used for the browser agents

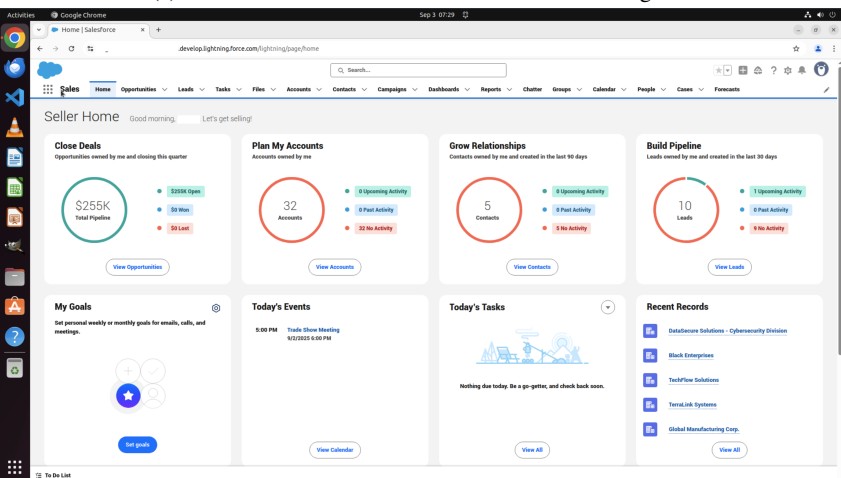

(b) Screenshot of the entire desktop used for the computer-use agents

Figure 6: Different types for screenshots.

---

**Sample of DOM texts**

```
[0]<a Skip to Navigation/>
[1]<a Skip to Main Content/> Setup
[2]<input text;combobox;false;Search Setup/> Search Setup
[3]<button Favorite this item>This item doesn't support favorites/>
```

---

```
[4]<button Favorites list/>
[5]<a ;button;false>Global Actions/>
[6]<button Guidance Center/>Guidance Center
[7]<button Salesforce Help/>
[8]<a ;button;false>Setup/>
[9]<button 3 Notifications/> 3 new notifications
[10]<button View profile/>
[11]<button false>App Launcher/> Setup
[12]<a tab>Home/>
[13]<a tab>Object Manager/>
[14]<a ;button;false>Object Manager List/> Quick Find
[15]<input Quick Find;search/>
[16]<li Setup Home;treeitem>Expand/>
[17]<a Setup Home/>
[18]<li Service Setup Assistant;treeitem>Expand/>
[19]<a Service Setup Assistant/>
[20]<li treeitem;Commerce Setup Assistant>Expand/>
<... omit for brevity ...>
```

## C  AGENT ACTION SPACE

### C.1  BROWSER-USE AGENTS

Browser-use agents is implemented via the `browser-use` framework with our extensive modification on the DOM parser, the action space is described in Table 4.

Table 4: The action space of the browser-use agents.

| Action | Description |
|---|---|
| **Primitive Actions** | |
| go_to_url(url: str, new_tab: bool) | Navigate to a given URL in the current or new tab depending on new_tab |
| go_back | Navigate back to the previous page |
| click_element_by_index(index: int, while_holding_ctrl: bool ) | Click an element at the given index; Ctrl+Click to open in new background tab |
| input_text(index: int, text: str, clear_existing: bool) | Input text into an element at the given index; optionally clear existing text first |
| switch_tab(tab_id: str) | Switch to the browser tab identified by tab_id |
| close_tab(tab_id: str) | Close the browser tab identified by the tab_id |
| scroll(down: bool, num_pages: float, frame_element_index: int ) | Scroll up/down a given number of pages, optionally inside a specific frame element |
| scroll_to_text(text:str) | Scroll to a text in the current page. |
| send_keys(keys: str) | Send keystrokes to the active element or page |
| get_dropdown_options(index: int) | Get all option values from a dropdown element at the given index |
| select_dropdown_option(index: int, text: str) | Select a dropdown option by its text or exact value |
| wait(seconds: int) | Wait for specified seconds. Can be used to wait until the page is fully loaded. |
| done(text: str, success: bool, files_to_display: list[str] ) | Signal that the task is done, with optional files to display |
| **File Manipulation** | |
| upload_file_to_element(index:int, path:str) | Upload file to interactive element with file path |
| write_file(file_name:str, content:str, append: bool) | Write or append content in file system. |
| replace_file_str(file_name: str, old_str: str, new_str:str) | Replace old_str with new_str in file_name |
| read_file(file_name:str) | Read file_name from file system |
| **Additional Toolboxes** | |
| search_google(query: str) | Perform a Google search with the given query |
| extract_structured_data(query:str, page_html:str) | Sends query and current page to an LLM to extract structured and semantic data. |

### C.2  COMPUTER-USE AGENTS

Our implementation is built upon and extends the implementation of `OSWorld` (Xie et al., 2024). Therefore, the action space of computer-use agents is the same as the one used in `OSWorld`. Please refer to Section A.3 in Xie et al. (2024) for details. We repeat it in Table 5 for completeness.

## D  CONSTRUCTION OF HUMAN-DEMONSTRATION

To construct the demonstration, we summarize the high-level plan with UI elements operations by distilling the human trajectories. We turn human trajectories into demonstration examples by prompting the `GPT-5` to summarize it into actionable plans. One example is provided below.

Table 5: The action space of the computer-use agents.

| Action | Description |
|---|---|
| **Primitive Actions** | |
| moveTo(x=X, y=Y, duration=num_seconds) | Move mouse to XY coordinates over num_seconds seconds. |
| dragTo(x=X, y=Y, duration=num_seconds) | Drag mouse to XY coordinates over num_seconds seconds. |
| click(x=X, y=Y, clicks=num_of_clicks, button='left') | Click at XY coordinates with specified number of clicks and mouse button. |
| rightClick(x=X, y=Y) | Right-click at XY coordinates. |
| middleClick(x=X, y=Y) | Middle-click at XY coordinates. |
| doubleClick(x=X, y=Y) | Double-click at XY coordinates. |
| tripleClick(x=X, y=Y) | Triple-click at XY coordinates. |
| scroll(amount_to_scroll, x=X, y=Y) | Scroll the mouse wheel by given amount at XY coordinates. |
| mouseDown(x=X, y=Y, button='left') | Press and hold a mouse button at XY coordinates. |
| mouseUp(x=X, y=Y, button='left') | Release a mouse button at XY coordinates. |
| typewrite(str, interval=delay_secs) | Type text with optional delay between keystrokes. |
| hotkey(keycomb) | Press a combination of keys simultaneously. |
| keyDown(key_name) | Hold down a key. |
| keyUp(key_name) | Release a key. |
| Done | Signal that the task is done |

The task prompt is *"Create a Queue with the name "Shoe Case Support". Only send email to members under the distribution list "support.shoe@papaltd.com". Add the Object "Customer". Include "Role: Customer Support, North America" as the Member.".*

```
Sample Human Trajectory

The task is successful. The trajectory is as follows:
STEP:1 Perform the action [click] on the element: Log In.
STEP:2 Perform the action [click] on the element: Setup.
STEP:3 Perform the action [click] on the element: Setup.
STEP:4 Perform the action [click] on the element: Users.
STEP:5 Perform the action [click] on the element: Queues.
STEP:6 Perform the action [click] on the element: New Queue.
STEP:7 Perform the action [write] on the element: Textbox for Queue
     Label with the text: Shoe Case Support.
STEP:8 Perform the action [write] on the element: Textbox for Queue
     Email with the text: support.shoe@papaltd.com.
STEP:9 Perform the action [scroll down].
STEP:10 Perform the action [click] on the element: Customer.
STEP:11 Perform the action [click] on the element: Add to Selected
     Objects.
STEP:12 Perform the action [write] on the element: Dropdown for
     Queue Members with the text: Role.
STEP:13 Perform the action [click] on the element: Role: Customer
     Support, North America.
STEP:14 Perform the action [click] on the element: Add to Selected
     Members.
STEP:15 Perform the action [click] on the element: Save.
```

```
Sample Demonstration Summarized from Human Trajectory

Successful plan:
- Log in and open Setup.
- Navigate to Users > Queues.
- Click New Queue.
- Configure the queue:
  - Queue Label: Shoe Case Support.
  - Queue Email: support.shoe@papaltd.com.
- Add supported object:
  - Select Customer and click Add to Selected Objects.
- Add members:
  - In Queue Members, choose Role, select Role: Customer Support,
     North America, and click Add to Selected Members.
- Save the queue.
```

For each template, we crafted two demonstration examples. As a result, two samples are tested with *exact* demonstration and three samples are tested with the *similar* demonstration. This setting can reveal two important abilities of the agents: 1) Among a given set of the demonstration, can the agent identify and follow the exact demonstration to finish the task? 2) Can the agent leverage similar successful demonstration to complete the current task? The results are presented in Table 6.

## E  METRICS BREAKDOWN FOR THE DEMONSTRATION-AUGMENTED SETTING

In the demonstration-augmented setting, we provide two demonstration examples. As a result, two task instances are tested with the exact demonstration examples but the agent still needs to figure out which one to use (corresponds to "w/ demo." column in Table 6). This simplifies the test by setting the recall (of the relevant demonstration) to 100%. The remaining three task instances are tested only with the similar demonstration examples (corresponds to "w/o demo." column in Table 6).

One can make the following observations.

- For task instances under the "w/o demo." category, the demonstration examples consistently improve milestone score, task success rate, and time consumption metrics. The steps, tokens, and costs metrics are also in general improved only with a few exceptions.

- For task instances under the "w/ demo." category, the demonstration examples helps with the methods' performance if their performance under the zero-shot setting is not high.

- Different models show different capabilities in utilizing the demonstration, with the closed-source models being significantly capable than the open-source models.

Table 6: Performance metrics breakdown. The percentage number within the parentheses is the change with respect to the zero-shot setting. All numbers are averaged across samples. We do not report the costs for self-served models. The costs for `Agent-S2.5(w/GPT-5)` does not account for the cost of serving `UI-TARS-7-B` as a grounding model.

| Method Name | Milestone Score ↑ | | Success Rate ↑ | | Time (min) ↓ | |
|---|---|---|---|---|---|---|
| | w/ demo. | w/o demo. | w/ demo. | w/o demo. | w/ demo. | w/o demo. |
| Browser-Use Agents | | | | | | |
| GPT-5 | 0.75 (-1.03%) | 0.76 (1.08%) | 0.52 (-1.27%) | 0.56 (8.77%) | 16.70 (-13.04%) | 19.19 (-4.42%) |
| o3 | 0.69 (1.01%) | 0.69 (7.55%) | 0.50 (8.82%) | 0.50 (14.29%) | 16.20 (-23.15%) | 18.20 (-19.23%) |
| Gemini-2.5-Pro | 0.72 (60.16%) | 0.70 (29.36%) | 0.47 (70.73%) | 0.45 (25.00%) | 9.76 (-61.39%) | 10.37 (-54.12%) |
| Claude-4-sonnet | 0.66 (16.30%) | 0.72 (12.34%) | 0.42 (26.00%) | 0.51 (26.67%) | 11.23 (1.01%) | 11.17 (-8.00%) |
| Computer-Use Agents | | | | | | |
| UI-TARS-1.5-7B | 0.27 (243.38%) | 0.21 (98.91%) | 0.13 (1800.00%) | 0.05 (100.00%) | 6.62 (5.62%) | 6.38 (-4.09%) |
| OpenCUA-7B | 0.13 (156.29%) | 0.11 (207.20%) | 0.05 (300.00%) | 0.05 ($\infty$) | 18.79 (-10.93%) | 18.94 (-7.68%) |
| OpenAI-CUA | 0.51 (59.94%) | 0.48 (54.01%) | 0.31 (64.29%) | 0.26 (52.63%) | 4.97 (-13.50%) | 5.71 (-8.58%) |
| Claude-4-sonnet(computer) | 0.66 (25.29%) | 0.69 (22.36%) | 0.48 (60.00%) | 0.47 (44.44%) | 7.08 (-13.11%) | 7.66 (-4.50%) |
| Agent-S2.5(w/GPT-5) | 0.56 (-8.90%) | 0.59 (3.29%) | 0.38 (-13.85%) | 0.44 (28.95%) | 24.18 (-2.61%) | 25.23 (-9.58%) |

| Method Name | Steps ↓ | | Tokens (k) ↓ | | Costs ($) ↓ | |
|---|---|---|---|---|---|---|
| | w/ demo. | w/o demo. | w/ demo. | w/o demo. | w/ demo. | w/o demo. |
| Browser-Use Agents | | | | | | |
| GPT-5 | 19.99 (-11.37%) | 23.14 (-4.36%) | 216.49 (-10.91%) | 254.60 (-1.03%) | 0.47 (-14.82%) | 0.57 (-2.35%) |
| o3 | 24.44 (-6.28%) | 26.54 (-7.42%) | 227.82 (-2.27%) | 248.45 (-1.92%) | 0.54 (-2.03%) | 0.59 (-1.98%) |
| Gemini-2.5-Pro | 18.95 (25.46%) | 20.19 (35.24%) | 172.28 (27.94%) | 183.65 (46.72%) | 0.30 (26.05%) | 0.33 (45.05%) |
| Claude-4-sonnet | 23.99 (-0.25%) | 23.82 (-6.44%) | 368.36 (3.54%) | 358.39 (-4.19%) | 1.21 (3.25%) | 1.18 (-4.63%) |
| Computer-Use Agents | | | | | | |
| UI-TARS-1.5-7B | 39.18 (7.32%) | 37.32 (-3.07%) | 213.83 (14.69%) | 212.05 (5.25%) | - | - |
| OpenCUA-7B | 33.56 (4.52%) | 33.60 (6.27%) | 158.77 (15.44%) | 159.56 (17.07%) | - | - |
| OpenAI-CUA | 25.34 (-18.16%) | 29.91 (-11.44%) | 159.29 (-16.53%) | 188.89 (-10.29%) | 0.49 (-16.64%) | 0.58 (-10.43%) |
| Claude-4-sonnet(computer) | 30.85 (-13.69%) | 33.66 (-5.89%) | 405.68 (-12.82%) | 450.68 (-3.00%) | 1.28 (-12.80%) | 1.42 (-3.08%) |
| Agent-S2.5(w/GPT-5) | 34.33 (0.75%) | 34.56 (-9.55%) | 620.41 (5.75%) | 634.11 (-7.58%) | 1.17 (1.91%) | 1.21 (-8.43%) |

## F  ADDITIONAL DETAILS ON THE ANNOTATION PROCESS

Each annotator first needs to pass the on-boarding tests in order to be qualified for annotation. This process includes learning the basic usage of the Salesforce platform, the annotation and review tools. The annotation tool uses an action space similar to that described in Table 4 (only with primitive

actions). Due to the action space design, we also used the DOM parser we developed for browser-agents to put bounding boxes with the number tags on interactive elements on the Salesforce web pages. In this way, annotators can use prescribed actions with element indices to finish the tasks. This design has two advantages: 1) the collected trajectories are free from redundant and noisy user actions; 2) ensure action space design is self-complete and sufficient for agents to use.

Once the annotator is qualified, it starts the annotation process by first reading the knowledge article to understand the intention of the task templates and correct execution steps. Once the annotation is completed, the trajectories are sent for review to ensure final quality. Any rejected trajectories are sent to different annotators for rework.

## G   EXTENDED IMPLEMENTATION DETAILS

**Agent Implementation.**   We primarily focus on testing two types of agents, Browser-Use agents and Computer-Use agents. The implementation of the agent loop is adapted from the implementation proposed in `browser-use` (Müller & Žunič, 2024) and `OSWorld` (Xie et al., 2024). We made significant changes in the DOM parser provided by the `browser-use` to improve the quality of detecting all interactive elements on the Salesforce Platform and handling Iframe and cross-origin contents, which results in better quality of SOM screenshots and DOM texts. Without these changes, agents run with `browser-use` framework would perform significantly worse compared to the number we reported in Table 3. We also modify the `OSWorld`'s agent implementation to better handle tasks from `SCUBA`, for example, system prompts.

For the prediction service, we either directly use the API services or serve the models with the vLLM (Kwon et al., 2023) whenever it's possible. If we self-served the model, we create 8 replicas of the vLLM online inference endpoints, each occupying 1 GPU. The load balancing is implemented in a round-robin fashion. There is only 1 exception with the `OpenCUA-7B` since there's no vLLM support yet. We tried different solutions and finally choose to serve the model with ray (Moritz et al., 2018) with inference engine from Huggingface's Transformers (Wolf et al., 2019). We created 8 model instances with each on one 1 GPU and the load balancing is handled by the ray.

**Asynchronous Parallel Inference Infrastructure.**   As mentioned in Section 3, `SCUBA` supports asynchronous parallel evaluation to reduce the overall evaluation time. The asynchronous support for browser-use agents is straightforward by spinning up a desired number of browsers in the headless mode and assigning each agent to a dedicated browser with one task. For computer-use agents, the asynchronous inference pipelines starts by creating a desired number of Docker Containers on a remote server with each container running a Ubuntu system. Whenever the environment is not in use, we immediately assign an agent with a task to it. The container first reverts the system to initial snapshot, and the agent runs with a fresh desktop. Since we have more tasks than agents, this allows the full utilization of the docker containers. In this paper, the remote server is a n2-standard-96 instance (96 vCPUs and 384 GB memory). If the agent is served with the vLLM, the full evaluation can be done in less than 90 minutes with 40 parallel containers.

**Metrics**   To give a full picture of the agents' performance, we design the following metrics, to cover the accuracy, efficiency, and costs, three arguably the most important aspects when developing the agents in the enterprise use cases. In the accuracy dimension, we use the Milestone Score (a number between 0 and 1) and Task Success(0 for failure and 1 for success), which is described in Section 3.2.2. In the latency dimension, we designed two metrics. The Time metric captures the wall clock time required to finish one task. This only captures the actual users' experience, but not the full picture. Therefore, we also use the metrics #Steps to capture more information. The #Steps metric counts the number of prediction made. The nuance is that with one prediction, some agents may predict multiple actions while the others may only predict one action at a time. This metric does not distinguish the difference. Lastly, we also report the costs to finish the task. We only provide this number for the paid API service. Specifically, for `OpenAI-CUA`, it's $3.00/M input tokens and $12.00/M output tokens; for `Claude-4-sonnet`, it's $3.00/M input tokens and $15.00/M output tokens; for `GPT-5`, it's $1.25/M input tokens and $10.00/M output tokens; for `o3`, it's $2.00/M input tokens and $8.00/M output tokens; for `Gemini-2.5-pro`, it's $1.25/M input tokens and $10.00/M output tokens. We also use #Tokens to record the total token consumption. The more token it consumes, the overall time would be proportionally longer and more costly.

# H  QUALITATIVE ANALYSIS

In this section, we provide qualitative analysis of the common failure modes and potential remedies.

**Observation 1: Grounding is still a persistent issue for the computer-use agents.**    In the example shown in Figure 7, the agent is supposed to click the "Add" button (right arrow). The `Claude-4-sonnet`(computer)'s thought is *"Perfect! Now I can see "Case" in the Available Objects list. Let me click on "Case" to select it"*. However, the predicted action is `click(378, 693)`, whose coordinate is on the text "Add" instead of the button (visualized as the red dot). On the contrary, when `Claude-4-sonnet` used with `browser-use`, it only needs to select the element id. Its prediction becomes `click_element(index=121)`, which is correct. One additional grounding failure is provided in Figure 8.

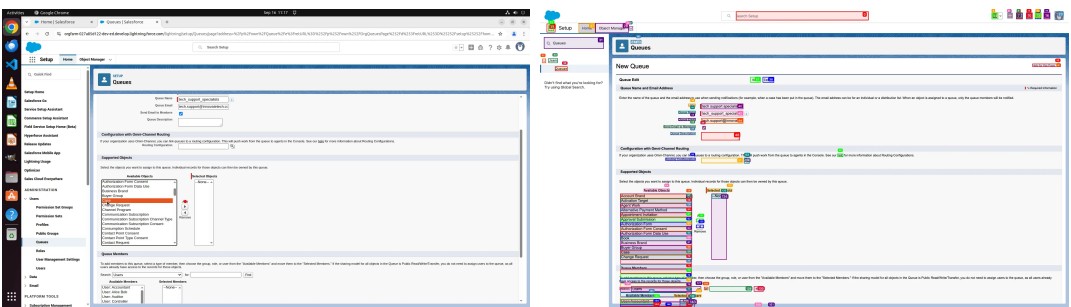

(a) Screenshot for `Claude-4-sonnet`(computer)     (b) Screenshot for `Claude-4-sonnet`(browser)

Figure 7: Different grounding difficulty.

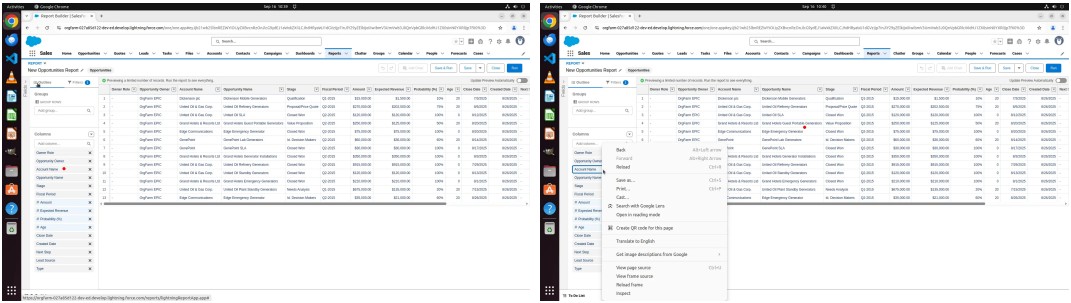

(a) Screenshot for `Claude-4-sonnet`(computer) on step 8     (b) Screenshot for `Claude-4-sonnet`(computer) on step 9

Figure 8: Agent selects the wrong cell.

**Potential solution.** To address the grounding issue for computer-use agents, one potential solution is to ask the models to make multiple predictions (with confidence) about the coordinates and do a majority voting or weighted average. For example, in the example shown Figure 9, at the step 27, the agent's thought is *"I noticed an "Edit" button at the top of the page, ..., it's important to first click this Edit button to enter the detailed settings page."* The initial predicted coordinate was wrong. After multiple attemps, the agent final get the coordinate right at the step 34. This idea is aligned with the recent work (Yang et al., 2025c). We also want to emphasize that the SOM is not the ultimate solution to address the grounding issue. The out-of-box DOM parser from `browser-use` cannot detect the add button (index 121) in Figure 9 (b), making the task impossible to be finished. Similar issues can happen with other websites and having a universal DOM parser that works for all websites is very challenging.

**Observation 2: Ineffective history managing strategy can mislead computer-use agents tracking progress.**    The history $H_t$ at the timestamp $t$, used for computer-use agents is the concatenation

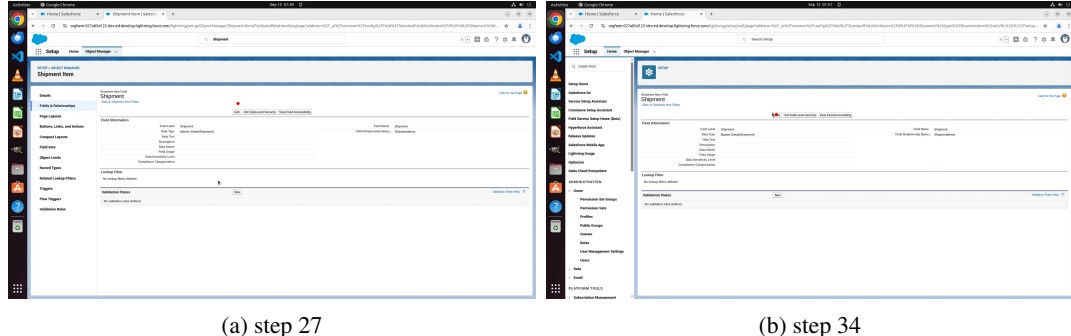

(a) step 27                                          (b) step 34

Figure 9: Multiple attempts can get the right coordinate.

of the past actions and the latest screenshot. This can be problematic and further hurts the task success rates when grounding is not accurate. Take the example from Figure 7 again to elaborate.

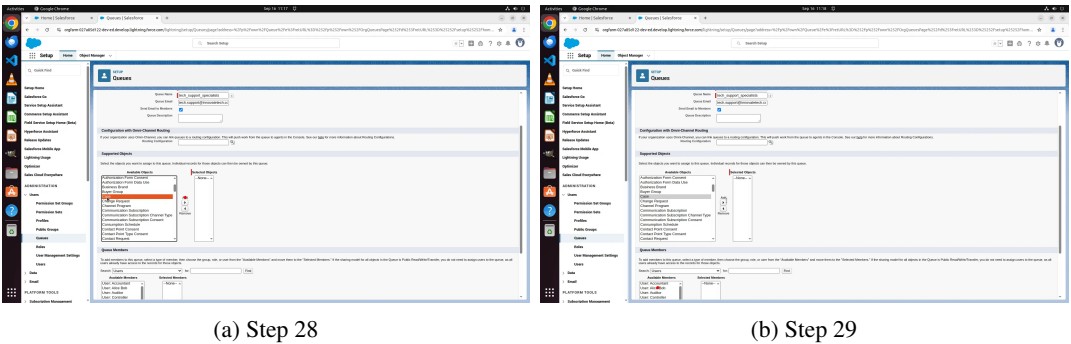

(a) Step 28                                          (b) Step 29

Figure 10: Incorrectly track the progress.

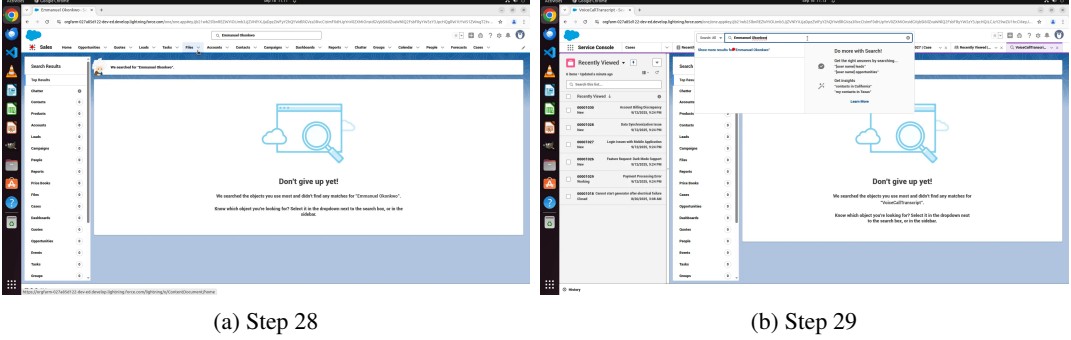

(a) Step 28                                          (b) Step 29

Figure 11: Additional example of incorrectly track the progress. The agent search the same object after 8 steps.

At the step 28, the agent failed to click the "Add" button. But the action `click(378, 693)` is added to the history of the step 29, therefore, the agent's thought became *"Perfect! I can see that "Case" is now highlighted in blue, and looking at the Selected Objects section on the right, I can see it no longer shows "–None–". Now I need to add Alice Bob as a member to the queue. Looking at the Queue Members section at the bottom, I can see "User: Alice Bob" in the Available Members list. Let me click on "User: Alice Bob" to select it."* (See Figure 10 (b)). We can see that even the agent did not successfully click the "Add" button, it thought the sub-task was done. Another example could be found in Figure 11, where the agent search for the same object in step 8 and step 16.

**Potential solution.** Reflection is an effective way to alleviate this phenomenon. `Agent-S2.5` also did not predict the coordinate correctly in this step. However, it has a reflection mechanism to output something like, *"(Previous action verification) Previous attempts to add "Case" via the Add button and double-click did not move it to Selected Objects. The Selected Objects list still shows "–None–."* With this reflection, the `Agent-S2.5` finally managed to click the "Add" button and complete the task successfully. `browser-use` also has the similar design to instruct the agent to evaluation if previous goal was achieved or not.

**Observation 3: Human Demonstration is correct yet not necessarily optimal.** Recall from Table 3 and Table 6, there are some cases the human demonstration also increase the latency and costs despite the performances are greatly improved. One reason is that the agent simply does not take enough steps to finish the tasks under the zero-shot setting. The increase in latency and costs can not be avoided. However, there are other cases where the human demonstration is not efficient and the agent can take shortcuts under the zero-shot setting.

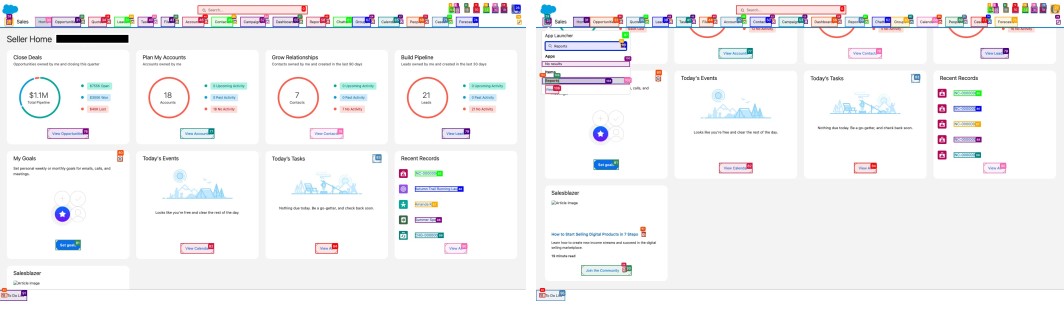

(a) zero-shot setting; Step 1        (b) demonstration-augmented setting; Step 4

Figure 12: Agent take different actions under the zero-shot and demonstration-augmented settings.

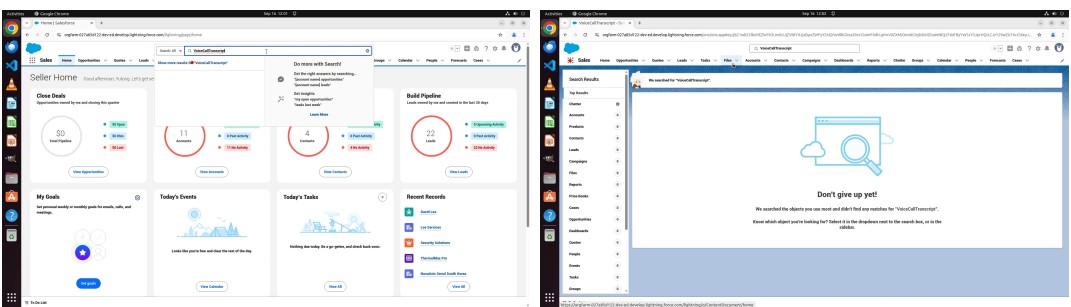

(a) Demonstration-augmented setting; Step 4        (b) Demonstration-augmented setting; Step 5

Figure 13: Agent still can not choose the correct searching box even with demonstration-augmented settings.

As shown in the Figure 12, the sub-task is to navigate to the "Report" page. Under the zero-shot setting, the agent issued the action `click_element(index=58)`, which clicks the "Report" tab. However, under the demonstration-augmented setting, the agent predicted a sequence of actions as `click_element(index=28)` → `input_text(index=98` → `text='Reports', click_element(index=105))` to achieve the same goal. This differences precisely originates from the human demonstration, where the annotator exactly did *"- Log in to Salesforce and open the App Launcher. - Search for and open Reports,"*. And it turned out that the annotator learned this from the knowledge article in the Trailhead website. Another example is presented in Figure 13.

**Potential solution.** This is an interesting research direction on how to adapt the knowledge article to the agent's performance. We leave this for future research.

**Observation 4: Agents can easily fall into navigation dead ends and fail to recover.** As illustrated in Figure 14 and Figure 15, even strong agents can accidentally enter irreversible states during multi-step computer-use tasks. In Figure 14, the agent unexpectedly launches a new browser window, thereby losing its previous login state and becoming unable to proceed with the remaining steps. Similarly, in Figure 15, the agent opens a new tab and subsequently closes the entire browser, leaving itself in an unrecoverable state at step 4. These behaviors reveal that current agents lack a robust mechanism for escape actions—they do not reliably detect when they are deviating from the expected workflow, nor do they possess the ability to navigate back to a safe state once such deviations occur. Unlike performance issues that stem from insufficient reasoning or grounding, dead-end states introduce a stricter form of failure: even with correct planning afterward, the task becomes impossible to complete due to earlier irreversible actions.

**Potential solution.** A promising direction is developing navigation-recoverability mechanisms for web agents. These include (1) state-change detection modules that can identify when the UI or browser state diverges from the intended trajectory, (2) explicit "rollback" or "safe-state restoration" policies that recover previous login states or reopen closed contexts, and (3) training demonstrations that emphasize recovery behaviors rather than only optimal trajectories. Designing such robust escape strategies is an important open challenge for building reliable, real-world computer-use agents, and we leave comprehensive exploration of this direction to future work.

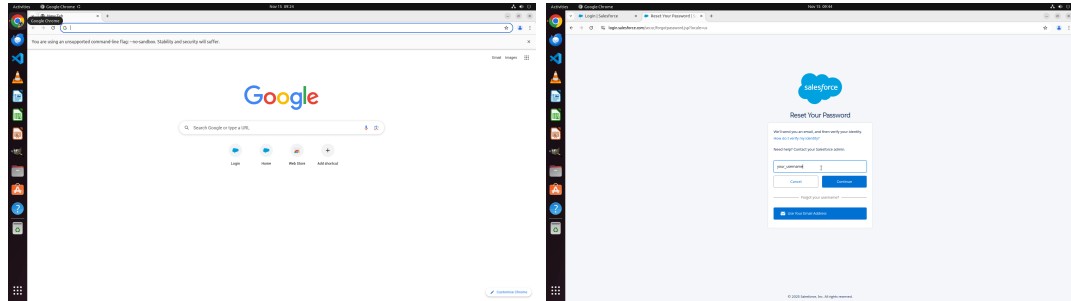

(a) The agent started a new browser and loses the login status, step 4

(b) The agent could not go back through the remaining steps, step 50

Figure 14: Example of dead end where the agent left the login status.

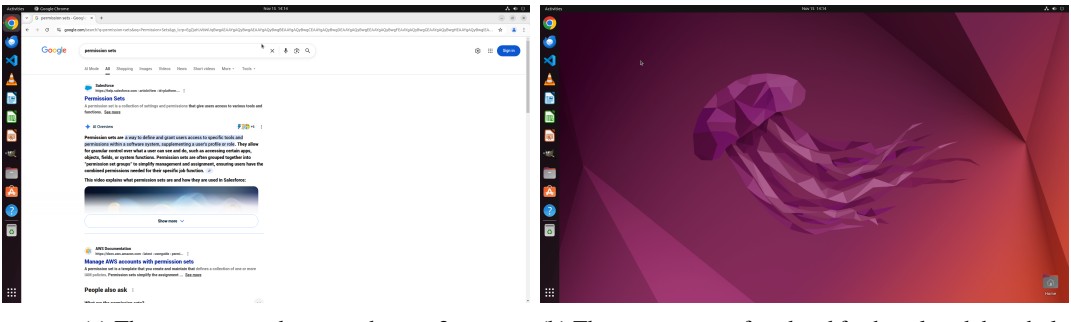

(a) The agent opened a new tab, step 2

(b) The agent got confused and further closed the whole browser, step 4

Figure 15: Example of dead end where the agent left the login status.

**Observation 5: Agents often lack essential domain knowledge to navigate enterprise environments.** As shown in Figure 16, even when the interface is clearly visible, the agent may fail to locate critical UI entry points due to missing domain-specific knowledge. In this example, the agent repeatedly searches for the Salesforce "Setup" page but still cannot identify the correct navigation path—even after 36 steps. This failure is not due to grounding or perception issues: the relevant UI elements are present, and the agent correctly recognizes the surrounding content. Instead, the agent lacks an a priori understanding of common enterprise platform conventions (e.g., Salesforce's

gear icon leading to Setup, object manager locations, admin navigation flows). Without such prior knowledge, the agent explores the interface in a trial-and-error manner, wasting many steps and often failing to reach the correct page at all. This highlights a fundamental gap between generic web-navigation ability and the specialized domain knowledge required for realistic enterprise tasks.

**Potential solution.** One direction is to incorporate domain-structured priors into the agent—e.g., curated Salesforce navigation knowledge, task-specific demonstrations, or symbolic UI maps that specify where essential admin features reside. Another promising approach is to pretrain or finetune on large-scale enterprise interaction logs so the agent internalizes common UI patterns. Overall, integrating domain knowledge remains a key challenge for scaling computer-use agents to real enterprise workflows.

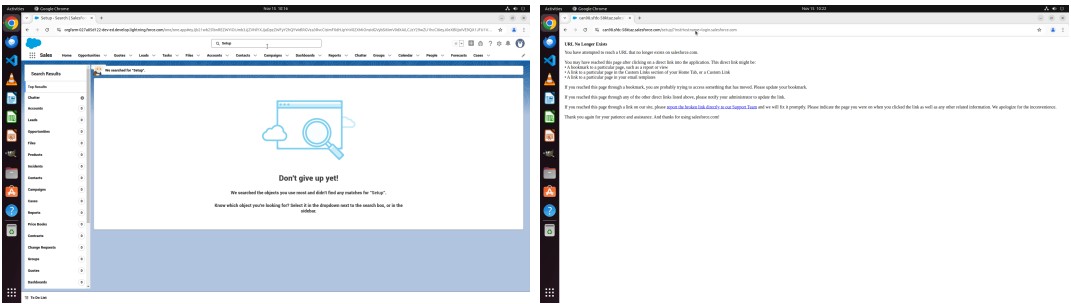

(a) The agent does not know where to open setup page in Salesforce website, step 10

(b) The agent still does not know where to open setup page in Salesforce website, step 36

Figure 16: Example of lack of Salesforce domain knowledge.

**Observation 6: Agents struggle to recover once an early mistake is made.** As shown in Figure 18, even a small early mistake can render the remainder of the trajectory ineffective. In this example, the agent accidentally saves incorrect form information at step 10, but instead of recognizing the inconsistency or attempting to revise the entry, it continues executing future steps under the assumption that the earlier action was correct. By step 50, the agent is still unable to correct the form, resulting in a failure despite otherwise reasonable planning. This showcases a common weakness in current computer-use agents: they lack a mechanism for error introspection—the ability to detect, revisit, and amend earlier steps when the downstream state diverges from the expected trajectory. Unlike grounding or navigation issues, this failure mode is particularly severe because the agent's plan may be logically correct, yet the initial incorrect state makes success impossible without rollback or self-correction.

**Potential solution.** A promising direction is to incorporate action-level verification and self-correction loops, enabling the agent to periodically re-evaluate whether prior actions produced the intended UI state. This could include explicit state-validation checkpoints, rollback strategies, or memory-based action traces that allow the agent to detect when its current state is inconsistent with the task goal. Developing reliable recovery mechanisms is crucial for enabling robust long-horizon enterprise tasks, and we consider this an important area for future work.

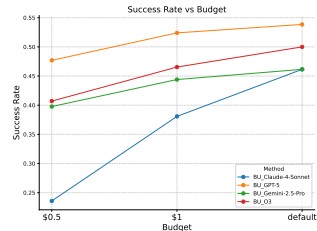

## I BUDGET-CONSTRAINED TEST SETTING

In this section, we consider a budget-constrained test setting to study different method's performance since cost is an important factor when deploying agents in the enterprise scenario. The results are shown in Figure 19. We chose browser-use agents with the demonstration setup for the testing purpose as it is the most performant setup according to Table 3. As shown in the Figure 19, all method's success rates are improved by allowing more budgets.

Figure 17: Budget limit per task on the final task success rates. The "default" means under the 50 steps budget used in Table 3.

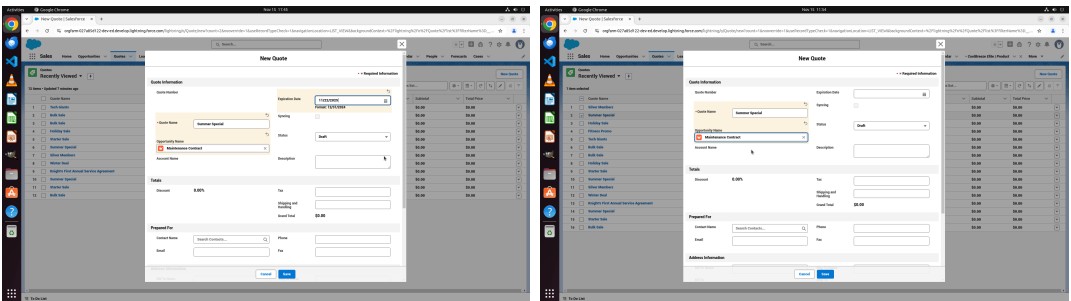

(a) The agent saved the wrong form information, step 10

(b) The agent was not able to correct the form until the end, step 50

Figure 18: Example of lack of Salesforce domain knowledge.

Figure 19: Performances of various agents on the SCUBA. All metrics are averaged accross all instances. ↑ next to each column name represents the larger value, the better and vice-versa for ↓.

| Method Name | Milestone Score ↑ | Success Rate ↑ | Time (min) ↓ | Steps ↓ | Tokens (k) ↓ | Costs ($) ↓ |
|---|---|---|---|---|---|---|
| **Demonstration-Augmented Setting with Browser-Use Agents** | | | | | | |
| budget $0.5 per task | | | | | | |
| GPT-5 | 0.65 | 47.69% | 9.64 | 14.93 | 125.98 | 0.30 |
| o3 | 0.60 | 40.70% | 10.02 | 18.06 | 139.09 | 0.33 |
| Gemini-2.5-Pro | 0.61 | 39.77% | 6.29 | 16.69 | 129.20 | 0.18 |
| Claude-4-sonnet | 0.41 | 23.85% | 5.52 | 13.73 | 149.01 | 0.45 |
| budget $1.0 per task | | | | | | |
| GPT-5 | 0.71 | 52.40% | 12.33 | 17.64 | 168.56 | 0.38 |
| o3 | 0.67 | 46.54% | 14.81 | 23.55 | 209.47 | 0.49 |
| Gemini-2.5-Pro | 0.67 | 44.40% | 7.54 | 20.24 | 171.22 | 0.23 |
| Claude-4-sonnet | 0.58 | 38.08% | 9.14 | 18.32 | 224.51 | 0.75 |

Judicious readers might notice that in Figure 19 the average costs of some agents are far below the budget limits. This is because some tasks require fewer steps than others, and even under the default budget setting (termination by 50 steps), the average costs also do not exceed $1. To support this claim, we visualize the distribution of costs per task for Claude-4-sonnet in Figure 20.

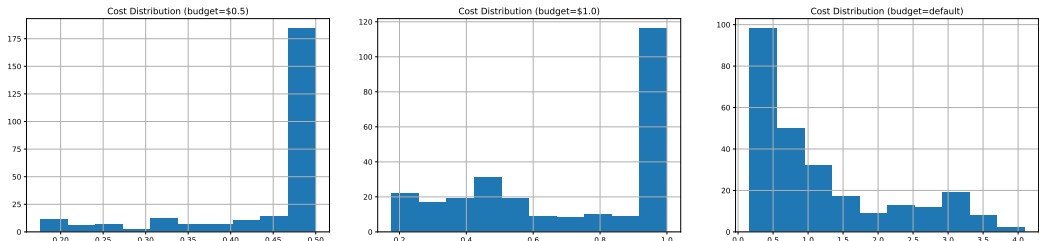

Figure 20: Cost Distribution per tasks for Claude-4-sonnet using browser-use framework.

