# OpenReview forum: "SCUBA: Salesforce Computer Use Benchmark"
_ICLR.cc/2026/Conference — ICLR 2026 Poster_

### Official Review · Reviewer_JVzK · 2025-10-20

**Soundness:** 2
**Presentation:** 3
**Contribution:** 2
**Rating:** 4
**Confidence:** 3

**Summary:**

This paper introduces a new benchmark for evaluating AI agents that use computers in only one website. The authors design a set of tasks, a rule-based evaluation system, and several case studies. The work involves notable engineering effort, but the paper lacks clear novelty and convincing evidence of realism.

**Strengths:**

The benchmark demonstrates notable implementation effort on a real web interface, reflecting practical interest in evaluating agent actions.

The paper clearly describes the task setup and examples, which helps readers understand the benchmark design.

The focus on web-based computer use provides a concrete starting point that could be extended to broader environments in the future.

**Weaknesses:**

The claim of “realistic” tasks is not well supported. Many prior benchmarks also make this claim without proper evidence.
Several important analyses and examples remain in the appendix. Some should be moved to the main text to support the main arguments.

The evaluation method is underexplained. The paper mentions rule-based evaluators but gives no clear description, pseudo-code, or examples. In Figure 2, the element ground_truth_dict appears, but its role in evaluation is unclear.

The rule-based evaluation raises safety concerns for automatic execution. It is unclear how potential side effects (e.g., unsafe file operations) are prevented.

The “Q&A troubleshooting” tasks look inconsistent with the main task types. Asking models to answer “yes or no” is unstable and not representative of real computer-use behavior.

The example with a 31-step task does not seem realistic. Long tasks like this are unreliable and should not be split into smaller tasks without showing how performance transfers.

**Questions:**

With rule-based evaluation, how to control side effects of the auto-execution. E.g. The model rm -rf all files before clicks a correct button? (maybe this is an extreme case)

Why use Q&A for troubleshooting as an additional type of agent execution？Maybe it should use some execution to find that why "yes" or "no" but not directly "only answer yes or no"?

Strong models like GPT and Claude pass near 50% of the tests, how to prove that the reasons of fails are the ability of the model or agent, but no an unreasonable task setting?

---

> ### Author Response · Authors · 2025-11-20
> **Official Comment by Authors (part 1)**
>
> We appreciate the reviewer's acknowledgment of the contribution of this work. We would like to address the reviewer's concerns as follows.
>
> > The claim of “realistic” tasks is not well supported. Many prior benchmarks also make this claim without proper evidence. Several important analyses and examples remain in the appendix. Some should be moved to the main text to support the main arguments.
>
> We thank the reviewer for the feedback.
>
> 1. Regarding the claim of “realistic” tasks: The tasks in SCUBA are directly sourced from real user interviews. As stated at the beginning of Section 3.2: *“This version of SCUBA contains 300 test cases, covering daily workflows from three types of personas, namely the platform administrator, the sales role, and the service role. The tasks are sourced by interviewing real Salesforce platform users, including solution engineers, org administrators, sales development representatives, business development representatives, and account executives. After collecting the use cases, we prepare the tasks by associating each with one knowledge article on Salesforce Trailhead, a learning platform that provides tutorials on how to use the Salesforce platform.”*
> We reiterate this here for the reviewer’s convenience. If the reviewer would like additional evidence to support this claim, we would be happy to provide it.
>
> 2. Regarding the arrangement of content: Due to page limitations, we aimed to balance detail and readability throughout the manuscript. We appreciate the reviewer’s suggestion and would be glad to move specific analyses or examples from the appendix to the main text if the reviewer can indicate which parts would be most helpful.
>
> > The evaluation method is underexplained. The paper mentions rule-based evaluators but gives no clear description, pseudo-code, or examples. In Figure 2, the element ground_truth_dict appears, but its role in evaluation is unclear.
>
> We thank the reviewer for their interest in knowing more details about the evaluators. Since most tasks from SCUBA require the agents to create and update the metadata and data records in the Salesforce platform, the evaluators are crafted to get the platforms's system states after the agent run.
>
> For example, in Figure 2, the task is to  *Set up a queue called "enterprise
> client support," associate it with the Case object, and include Alice Bob and Sofia Bennett as members.*  Then the corresponding evaluator contains a SOQL query, which is a Salesforce platform-specific and SQL-like query, to check if the `queue` is created with the correct name `enterprise client support`, associated with the `Case` object, and the members are correctly added. After the execution of the SOQL query, one obtains some retrieved results, and they are compared to the `ground_truth_dict` to check if the agent gets the task done.
>
> For demonstration purposes, we attach a snippet of the SOQL queries used for this task.
>
> ```sql
> "SELECT Id, DoesSendEmailToMembers, Email FROM Group WHERE Type = 'Queue' AND DeveloperName = '{queue_name}'"
> "SELECT Id, Name, Type, RelatedId FROM Group WHERE Id IN (SELECT UserOrGroupId FROM GroupMember WHERE GroupId = '{retrieved_results['queue_check'].records[0]['Id']}')"
> ```
>
> Correspondingly, the following fields from the `ground_truth_dict`
>
> ```json
> "Queue_name": "enterprise_client_support",
> "role_members": ["Alice Bob","Sofia Bennett"],
> ```
>
> are used to check if the retrieved results match. We will open-source this benchmark, so more details can be found in the actual evaluator implementation.
>
> > The rule-based evaluation raises safety concerns for automatic execution. It is unclear how potential side effects (e.g., unsafe file operations) are prevented.
> > With rule-based evaluation, how to control side effects of the auto-execution. E.g. The model rm -rf all files before clicks a correct button? (maybe this is an extreme case)
>
> We thank the reviewer for raising this important concern.
>
> The rule-based evaluators are human-crafted and do not contain any malicious/dangerous code, such as `rm -rf`. As mentioned in response to a previous question, all code-based evaluators are read-only, used solely to retrieve data and metadata. The agent never has access to the evaluator code and cannot execute it.
>
> We do understand the reviewer’s concern regarding data integrity, since the agent can manipulate data via UI operations. First, this concern is independent of the rule-based evaluation. Second, to ensure data integrity, the environment is reset to its initial state before each agent run, preventing any unintended changes or deletions.

---

> ### Author Response · Authors · 2025-11-20
> **Official Comment by Authors (part 2)**
>
> > The “Q&A troubleshooting” tasks look inconsistent with the main task types. Asking models to answer “yes or no” is unstable and not representative of real computer-use behavior.
> > Why use Q&A for troubleshooting as an additional type of agent execution?Maybe it should use some execution to find that why "yes" or "no" but not directly "only answer yes or no"?
>
> Q&A tasks in SCUBA require the agent to navigate to certain pages or locate certain data records to answer the questions, i.e., all answers are grounded on the Salesforce Platform's actual metadata and data records.  Some sample tasks are provided below to support our explanation.
>
> 1. Grounded on the Salesforce org's Opportunity object, which Profile has the most limited access. Choose one and reply with the exact name from the list [Custom: Sales Profile, Gold Partner User, Analytics Cloud Integration User].
> 2. Can the lead Ali Hussein be qualified based on the latest discussions? If the answer is no, which factor ('Budget', 'Authority', 'Need', or 'Timeline') is responsible? Return only one factor that the lead qualification fails to meet (i.e. 'Budget', 'Authority', 'Need', 'Timeline'). If the lead is qualified, return "Qualified".
>
> Therefore, these tasks precisely meet the reviewer's expectation, i.e., using some execution to find why "yes" or "no". In addition, our Q&A tasks go beyond yes-and-no type of questions as shown in the example above.
>
> > The example with a 31-step task does not seem realistic. Long tasks like this are unreliable and should not be split into smaller tasks without showing how performance transfers.
>
> We respectfully argue that we do not see why requiring 31 steps would make the task unrealistic. In fact, the average number of steps for human annotators exceeds 20, reflecting the complexity of enterprise workflows. For clarification, each UI action in the action space is counted as one step.
> For the task:
> “Create a territory model with the name ‘by Finance Industry’ and with the rule name ‘Industry, Active, and Type’. The selection criteria are ‘Industry equals Finance’, ‘Account equals Active’, and ‘Type is not equal to Channel/Partner Reseller or Installation Partner’. If the sales territory is not enabled, enable it first.”
> It requires 31 steps because humans must navigate five levels of page depth to locate the configuration area, and setting up the details of the territory model’s data schema in Salesforce requires more than 20 UI actions. This illustrates the realistic complexity of enterprise tasks.
>
> > Strong models like GPT and Claude pass near 50% of the tests, how to prove that the reasons of fails are the ability of the model or agent, but no an unreasonable task setting?
>
> All tasks are annotated by humans using the same action space available to the agents, and the annotated trajectories are further reviewed by humans to ensure correctness. Therefore, we are confident that the tasks are reasonable and that failures reflect the abilities of the agents, rather than any issues with the task design and settings.
>
>
> ---
>
> We sincerely thank the reviewer for the thoughtful feedback and for engaging closely with our work. We have carefully addressed all raised concerns. We hope that the additional explanations help better convey the contributions of our work.
>
> If the reviewer finds that these clarifications strengthen the paper and resolve earlier concerns, we would greatly appreciate a reconsideration of the overall assessment and score.
>
> We remain very open to further suggestions that could help improve the paper. Thank you again for your time and constructive review.

---

> > ### Comment · Reviewer_JVzK · 2025-11-23
> >
> > Thank you to the authors for the detailed and thoughtful rebuttal. Many of my earlier questions (e.g., about task sourcing, evaluator implementation details, and the overall setup) have been addressed clearly, and I appreciate the effort that went into these clarifications.
> >
> > ### Side effects and rule-based evaluation
> >
> > My original concern was not about the safety of the evaluator code itself, but about agent behavior that achieves the goal while introducing undesirable side effects. As far as I can tell, the current rule-based evaluators check only whether the final system state matches a ground_truth_dict. They do not distinguish between a “clean” solution and one that, for example, deletes or alters unrelated data before arriving at the correct final state.
> >
> > This means the benchmark cannot really tell apart “reliable” behavior from “unsafe but coincidentally goal-satisfying” behavior. For a computer-use benchmark, this is a substantive limitation: it weakens the claim that the evaluation reflects the quality of agent behavior rather than just whether the goal condition eventually holds.
> >
> > ### From human-realistic to AI-reasonable task design (including Q&A)
> >
> > I appreciate the clarification that tasks are derived from real user workflows. This supports the claim that tasks are human-realistic. However, realism for humans does not automatically imply that the tasks are well-posed or optimally structured for AI agents.
> >
> > - For long workflows (e.g., 20–30+ step tasks), a human process can almost always be decomposed into smaller, more modular sub-tasks. Without such decomposition or an analysis of how sub-task performance transfers to the full workflow, long-horizon tasks become hard to interpret: failures may come from instruction ambiguity, UI quirks, memory limits, or other factors that remain entangled. This is a common issue in benchmarks “extracted from human workflows,” and I do not think this work fully resolves it.
> > - A related instance of this issue appears in the Q&A troubleshooting tasks. Although the rebuttal makes it clear that these questions are grounded in the Salesforce org and do require navigation, their surface form still closely resembles standard NLP QA prompts (“Is lead X qualified?”, “Which profile has the most limited access?”, etc.). Large language models are by default trained to answer such questions directly, not to first “go and look” in the environment unless the prompt and evaluation strongly enforce this behavior.
> >
> > In other words, the general concern is about the form in which human workflows are turned into AI tasks:
> >
> > - Are long workflows structured in a way that makes errors interpretable and performance attributable to specific capabilities?
> > - For Q&A-style tasks in a computer-use benchmark, do the prompts and evaluation reliably enforce an “execution-first, answer-second” pattern, or can models often answer from internal knowledge without genuine interaction?
> >
> > The rebuttal explains that Q&A tasks are grounded and require navigation in principle, but it does not concretely address prompt sensitivity or how the setup prevents them from effectively degrading into ordinary language QA under slightly different prompting.
> >
> > Overall, while the rebuttal significantly improves clarity on many aspects of the work, these remaining issues around side-effect-insensitive evaluation and the gap between “human-realistic” and “AI-reasonable” task and prompt design are important to me.

---

> > > ### Author Response · Authors · 2025-11-24
> > > **Follow-up Response by Authors (part 1)**
> > >
> > > We appreciate the reviewer's thoughtful comments, and some of them provoke our deep thoughts.
> > >
> > > > Concerns about the agent behavior that achieves the goal while introducing undesirable side effects
> > >
> > > This is a genuinely thought-provoking and critically important question, and worth addressing. To share our perspective, we’ll begin by reviewing current benchmarking practices, then outline our approach, and finally explore potential enhancements.
> > >
> > > **Types of evaluators in the computer-use benchmarks**. To our best knowledge, there are mainly two types of evaluators developed in the computer-use benchmarks. The first type is the rule-based evaluators, e.g., OSWorld[1] and SCUBA, where the manually-crafted code snippets are executed to validate whether the task is successful. The second type is using the LLM as a judge[2, 3].
> > >
> > > Both types of evaluators only check whether the task goal is eventually satisfied. We provide examples from [1] and [2] below.
> > >
> > > ```
> > > # evaluator from the OSWorld
> > >
> > > "instruction": "I want to make the tabs wrapped over multiple lines when exceeding available space, please help modify the setting of VS Code."
> > >
> > > "evaluator": {
> > >     "func": "check_json_settings",
> > >     "expected": {
> > >       "type": "rule",
> > >       "rules": {
> > >         "expected": {
> > >           "workbench.editor.wrapTabs": true
> > >         }
> > >       }
> > >     },
> > >     <...omit for brevity>
> > >   }
> > >
> > > def check_json_settings(actual: str, expected: str, **options) -> float:
> > >     <...omit for brevity>
> > >
> > >     try:
> > >         with open(actual, 'r') as f:
> > >             data = json.load(f)
> > >     except Exception as e:
> > >         return 0.0
> > >
> > >     expect = expected['expected']
> > >
> > >     # Check if all expected key-value pairs are in the actual data
> > >     for key, value in expect.items():
> > >         if key not in data or data[key] != value:
> > >             return 0.0
> > >
> > >     return 1.0
> > > ```
> > >
> > > ```
> > > # system prompt used for identifying key-points, which is
> > > # used in WebJudge proposed in [2].
> > > # WebJudge leverages key-points to test whether a task is done or not
> > >
> > > <...omit for brevity>
> > > **Objective**: Carefully analyze the task description and extract the critical elements explicitly mentioned in the task for achieving its goal.
> > >
> > > **Instructions**:
> > > 1. Read the task description carefully.
> > > 2. Identify and extract **key points** directly stated in the task description.
> > >    - A **key point** is a critical element, condition, or step explicitly mentioned in the task description.
> > >    - Do not infer or add any unstated elements.
> > > <...omit for brevity>
> > >
> > > **Respond with**:
> > > - **Key Points**: A numbered list of the explicit key points for completing this task, one per line, without explanations or additional details.
> > > ```
> > >
> > > So, we agree that, **like related benchmarks, SCUBA also may not fully capture whether the agent trajectory is "clean", i.e., does not modify task-irrelevant environment states.**
> > >
> > > [1] OSWorld: Benchmarking Multimodal Agents for Open-Ended Tasks in Real Computer Environments
> > >
> > > [2] An Illusion of Progress? Assessing the Current State of Web Agents
> > >
> > > [3] WebVoyager: Building an End-to-End Web Agent with Large Multimodal Models
> > >
> > > **Our approach.** When building SCUBA, we also asked ourselves questions like below.
> > >
> > > 1. What if the agent modifies/deletes the task-irrelevant data, and how can we detect them?
> > > 2. Suppose there are two records `A` and `A'`, and they have a similar name. The agent is supposed to update `A`, but what if the agent actually operates on `A'`? How to ensure the data integrity of `A'`?
> > >
> > > These questions are related to the safety concerns and echo with the "clean" trajectory requirement mentioned by the reviewer.
> > >
> > > Ideally, the robust approach to detecting the agent's unexpected or sneaky behavior in SCUBA is to compare the environment states before and after the agent runs. However, this approach is prohibitive since the environment state files are huge for the Salesforce Platform, and we can not download all of them even depleting the daily API quota. So, comparing the before-and-after environments' state approach is not feasible.

---

> > > > ### Author Response · Authors · 2025-11-24
> > > > **Follow-up Response by Authors (part 2)**
> > > >
> > > > (...continued...)
> > > >
> > > > A compromised solution is to **understand failure modes of the agents and add case-by-case detection**. We have the annotators to inspect trajectories generated by a subset of agents, namely, GPT-5, o3, Gemini-2.5-Pro, and Claude-4-sonnet. We instructed the annotators to inspect 1) whether the evaluators successfully capture whether the task is completed, and **2) whether the agents made any unintended changes to the environment, like adding data not asked for or accidentally deleting irrelevant data**.
> > > >
> > > > We indeed observed that the agent may **accidentally alter some relevant data fields that the task does not ask for**. Let's use the example in our previous response to illustrate: *Set up a queue called "enterprise client support," associate it with the Case object, and include Alice Bob and Sofia Bennett as members.*
> > > >
> > > > In the trajectory, we found that some agents may also come up with a fake email address when creating the `queue` data object, since the UI allows the user to configure the `queue` data with more than the name field.
> > > >
> > > > Our solution, in this example, is to check additional fields that are related to the `queue` object, e.g.,
> > > >
> > > > ```
> > > > "SELECT Id, DoesSendEmailToMembers, Email FROM Group WHERE Type = 'Queue' AND DeveloperName = '{queue_name}'".
> > > > ```
> > > >
> > > > **Admittedly, this is neither scalable nor comprehensive. But this is our attempt to approach this problem.**
> > > >
> > > > **Potential solutions.** We think that one may consider the hybrid approach when building the next generation of evaluators, i.e., using rule-based approach to check if the task is done and using LLM-as-a-judge to check if the trajectory is clean.

---

> ### Author Response · Authors · 2025-11-24
> **Follow-up Response by Authors (part 3)**
>
> > Are long workflows structured in a way that makes errors interpretable and performance attributable to specific capabilities?
>
> We would like to clarify that each task template corresponds to one core feature offered by the Salesforce Platform. For example, a user workflow might look like first creating a custom object (similar to the concept of a small database), then preparing the data records for this custom object, and lastly creating a report to visualize the data distribution. Consequently, we have three templates in SCUBA, relating to custom object creation, data records creation, and visualization report generation, respectively. Therefore, the "long-workflows" in SCUBA are already atomic in the sense that each template by itself tests whether an agent can use one core feature offered by the Salesforce Platform, and some templates can be combined to compose a more complex task.
>
> In other words, we construct tasks by the minimal features instead of by the number of UI actions required, since we find the former is more coherent and natural.
>
> > For Q&A-style tasks in a computer-use benchmark, do the prompts and evaluation reliably enforce an “execution-first, answer-second” pattern, or can models often answer from internal knowledge without genuine interaction?
>
> This is a great question, and the concern is very valid. We include Q&A style tasks because it is one of the most-wanted capabilities according to the user survey, i.e., grounded on the Salesforce Platform's data to troubleshoot the misconfiguration or find data records buried in the database, or answer questions based on a particular customer profile.
>
> At first glance, this type of need should be better addressed by RAG or the search agents. But we found that searching for a particular data record using the API is not intuitive in the Salesforce Platform unless you know the data record's internal ID  (something that looks like KJD096723), whereas the UI search offers more intuitive solutions and user experience.
>
> As for whether the agent would answer the question without navigation, we checked all agents' trajectories, and **empirically** found that all agents follow the “execution-first, answer-second” pattern. This can be attributed to the allowed action space for agents (please see Appendix C). Nearly all of them are execution-oriented, i.e, click and type. Only one action `done(text)` allows the agent to return the answer to the question. Therefore, all trajectories for Q&A-style tasks take more than 5 steps.
>
> As for the prompt sensitivity, we would like to provide some examples to illustrate that even when the user does not explicitly use the wording "grounded-on", the agent would still try to take actions first (since all trajectories take more than 5 steps).
>
> 1. Can the lead Ali Hussein be qualified based on the latest discussions? If the answer is no, which factor ('Budget', 'Authority', 'Need', or 'Timeline') is responsible? Return only one factor that the lead qualification fails to meet (i.e. 'Budget', 'Authority', 'Need', 'Timeline'). If the lead is qualified, return "Qualified".
>
> 2. I'm reviewing the call transcripts to extract useful information. Please examine the call between Carlos Nunez and Hiro Tanaka, and determine how many years the warranty lasts for the product that Carlos offers Hiro. Respond with the number only.
>
>
> **Again, we thank the reviewer for the insightful questions. We hope the response can clear the remaining concerns on the quality and contribution of this work, and we are happy to take more questions. We would appreciate it if the reviewer would consider updating the score in light of our response.**

---

### Official Review · Reviewer_Hddk · 2025-10-29

**Soundness:** 3
**Presentation:** 3
**Contribution:** 4
**Rating:** 6
**Confidence:** 3

**Summary:**

The paper presents SCUBA, a benchmark for evaluating computer-use agents on enterprise-grade CRM workflows within the Salesforce platform. Unlike prior browser-oriented or service-specific benchmarks, SCUBA emphasizes realism through sandboxed Salesforce environments, task instances sourced from user interviews across administrator/sales/service personas, and fine-grained rule-based evaluation with milestone scores. The authors benchmark a wide range of representative agents under zero-shot and demonstration-augmented settings.

**Strengths:**

Tasks are grounded in real enterprise workflows and executed in genuine Salesforce sandboxes rather than synthetic mockups, addressing a major gap left by OSWorld/WebArena-like benchmarks.

The inclusion of process rewards, latency and cost metrics, and rule-based evaluators provides more diagnostic insight than binary success metrics commonly used in prior work.

**Weaknesses:**

1. As acknowledged, benchmark stability and long-term reproducibility depend on third-party platform updates, which may undermine comparability over time. Is it possible to provide a version-frozen evaluation environment to improve the reusability of the results?

2. The paper reports cost/time for agent inference, but does not quantify the annotation cost or the engineering overhead associated with constructing and maintaining SCUBA.

3. The reported improvements (e.g., from demonstration augmentation) are presented as absolute numbers. Could the authors conduct a statistical analysis to verify the robustness of these gains?

**Questions:**

Please see the weakness.

---

> ### Author Response · Authors · 2025-11-20
> **Official Comment by Authors**
>
> We appreciate the reviewer's acknowledgment of the contribution of this work, i.e., real environment, enterprise-level tasks, and fine-grained evaluators.  We would like to address the reviewer's concerns as follows.
>
> > As acknowledged, benchmark stability and long-term reproducibility depend on third-party platform updates, which may undermine comparability over time. Is it possible to provide a version-frozen evaluation environment to improve the reusability of the results?
>
> We explored the possibility of creating a version-frozen Salesforce environment and reached out to the Salesforce support team, but unfortunately, such an option is not available. This limitation is not unique to our work. Benchmarks such as online-Mind2Web[1] and WebVoyager[2] face the same constraint because they also depend on evolving live websites.
>
> However, as we replied to the reviewer **QXNn** and **X9Qy**, we will **commit to long-term maintenance and update of this benchmark**.
>
> Beyond these human efforts, several factors help ensure that task validity can persist for a substantial period. Although Salesforce regularly updates its APIs, the platform provides strong backward compatibility guarantees. In addition, when designing the tasks, we deliberately focused on workflows built on features that have been stable and well-supported for many years. Conversely, we intentionally excluded tasks involving rapidly evolving capabilities, such as the new Agentforce features, because their UI and APIs change too quickly to reliably maintain in a benchmark.
>
> These considerations collectively help mitigate platform change and enhance the long-term reusability of SCUBA.
>
> [1] An Illusion of Progress? Assessing the Current State of Web Agents
> [2] WebVoyager: Building an End-to-End Web Agent with Large Multimodal Models
>
> > The paper reports cost/time for agent inference, but does not quantify the annotation cost or the engineering overhead associated with constructing and maintaining SCUBA.
>
> This is a valuable point. Building SCUBA required substantial human and engineering effort. Approximately two people spent around three months developing the tasks, crafting the evaluators, configuring the environment, and constructing the agent loop. In addition, three annotators collectively spent about one month on all task annotation and quality assurance.
>
> At present, the ongoing maintenance cost is relatively low, largely because Salesforce provides strong API backward compatibility, which minimizes the required updates. However, we remain committed to revisiting and updating the benchmark as needed to ensure its long-term reliability.
>
>
> > The reported improvements (e.g., from demonstration augmentation) are presented as absolute numbers. Could the authors conduct a statistical analysis to verify the robustness of these gains?
>
> We appreciate this suggestion. We will attempt to report means and standard deviations if we can secure additional budget for paid LLM API service and computation resources. We agree that statistical analysis would strengthen the scientific rigor of the results.
>
> That said, we note that even when running the same agent on the same task twice, the outcomes often differ due to the long-horizon nature of the tasks and the inherent non-determinism of both the agents and the environment. Ideally, achieving statistically meaningful estimates would require averaging results over a large number of runs. While this is computationally expensive, we acknowledge that such analysis would provide a more robust characterization of performance variability and improvements.

---

### Official Review · Reviewer_YpFg · 2025-11-01

**Soundness:** 3
**Presentation:** 2
**Contribution:** 2
**Rating:** 2
**Confidence:** 4

**Summary:**

This paper introduces SCUBA (Salesforce Computer Use Benchmark), a new benchmark designed to evaluate computer-use agents on Customer Relationship Management workflows within the Salesforce platform. SCUBA contains 300 task instances derived from real user interviews, covering three primary personas: platform administrators, sales representatives, and service agents. It tests a wide range of enterprise-critical abilities, including UI navigation, data manipulation, workflow automation, information retrieval, and troubleshooting.

**Strengths:**

1. The benchmark runs in live Salesforce sandbox environments and uses fine-grained evaluation metrics that capture milestone progress, not just final success. This indicate some progress, as typical benchmarks like OSWorld[1] and ScienceBoard[2] only provide a 0/1 final success.
2. Tasks are sourced from real user interviews, ensuring practical relevance and representativeness. The evaluation is conducted within a live Salesforce sandbox env, offering higher fidelity than simulated platforms.
3. The study systematically evaluates both zero-shot performance and a "demonstration-augmented" setting. It empirically validates that providing demonstrations is an effective strategy.

[1] OSWorld: Benchmarking Multimodal Agents for Open-Ended Tasks in Real Computer Environments https://arxiv.org/abs/2404.07972

[2] ScienceBoard: Evaluating Multimodal Autonomous Agents in Realistic Scientific Workflows https://arxiv.org/abs/2505.19897

**Weaknesses:**

1. While advances have been made in the eval metrics, the SCUBA's infrastructure is wholly dependent on the Salesforce "sandbox" environment. This constitutes a significant threat to the long-term validity and reproducibility of the benchmark and imposes a heavy burden on the maintainers.
2. The current scale is relatively limited for capturing the vast complexity of an enterprise platform like Salesforce. Furthermore, existing works like AndroidWorld[3] and ScienceBoard[2] commonly employ randomized parameters to initialize the scenarios faced by the scaled agent tasks.
3. The task distribution is heavily skewed (57% for the 'admin'), which may limit the benchmark's ability to evaluate agent performance comprehensively across all core business roles.
4. For the experimens, the claim “The high task success rates seen in browser-use agents can be attributed to the combination of the stronger planning ability of the foundation models, richer observation space, and the more efficient action space design used in browser-use agents". This comparison between "Browser-Use" and "Computer-Use" agents seems to be methodologically unbalanced.
    a. The browser agents (or cuas leveraging a11ytrees [6]) receive a more structured observation space (SOM + DOM texts), which inherently poses no challenge for grounding.
    b. In contrast, other computer-using agents (e.g., SeeClick[3], OS-Atlas[4], UI-TARS[5]) take raw screenshots, which necessitates explicit grounding.
5. Minor: SCUBA is deeply coupled with Salesforce. While this provides depth in specific business scenarios, it sacrifices breadth. Conclusions drawn from this benchmark regarding UI nav and DOM-related stuffs may be difficult to generalize directly to other platforms.

[3] AndroidWorld: A Dynamic Benchmarking Environment for Autonomous Agents

[4] SeeClick: Harnessing GUI Grounding for Advanced Visual GUI Agents, ACL 24

[5] OS-Atlas: A Foundation Action Model For Generalist GUI Agents

[6] UI-TARS: Pioneering Automated GUI Interaction with Native Agents

[7] OS-Genesis: Automating GUI Agent Trajectory Construction via Reverse Task Synthesis https://arxiv.org/abs/2412.19723

**Questions:**

1. The paper highlights a dramatic performance drop for agents moving from OSWorld to SCUBA, attributing it to planning / grounding capabilities. This analysis could be deeper. Does this failure simply indicate a domain shift, or does it suggest that benchmarks for general desktop applications (like OSWorld) are poor proxies for the challenges of complex, web-based, stateful enterprise applications?

---

> ### Author Response · Authors · 2025-11-20
> **Official Comment by Authors (patr 1)**
>
> We appreciate the reviewer's acknowledgment of the contribution of this work, i.e., real environment, enterprise-level tasks, and fine-grained evaluators.  We would like to address the reviewer's concerns as follows.
>
> > While advances have been made in the eval metrics, SCUBA's infrastructure is wholly dependent on the Salesforce "sandbox" environment. This constitutes a significant threat to the long-term validity and reproducibility of the benchmark and imposes a heavy burden on the maintainers.
>
> We agree that maintaining task validity and evaluator reliability in a live benchmark is challenging. As we mentioned in our responses to reviewers **QXNn** and **X9Qy**, we are committed to the long-term maintenance and continuous improvement of this benchmark, and we hope to collaborate with the community to make it more accessible and robust.
>
> However, we respectfully disagree that the value of this work is diminished simply because it is a live benchmark running on the Salesforce sandbox environment. First, reviewers **QXNn**, **X9Qy**, and **Hddk** explicitly recognize the importance of SCUBA being grounded in a real environment. Second, several existing live benchmarks, such as online-Mind2Web[1] and WebVoyager[2], also define tasks on real websites. Without a live environment, it is difficult to meaningfully evaluate an agent’s true performance in production settings. We take a step further than [1] and [2] by manually crafting the evaluators using a rule-based approach to ensure unbiased evaluation and reproducibility. Third, seminal benchmarks like OSWorld continue to update their evaluators even though many of their tasks are relatively “static,” and this has not prevented OSWorld from becoming widely adopted in the CUA community.
>
> [1] An Illusion of Progress? Assessing the Current State of Web Agents
> [2] WebVoyager: Building an End-to-End Web Agent with Large Multimodal Models
>
>
> > The current scale is relatively limited for capturing the vast complexity of an enterprise platform like Salesforce. Furthermore, existing works like AndroidWorld[3] and ScienceBoard[2] commonly employ randomized parameters to initialize the scenarios faced by the scaled agent tasks.
>
> Constructing large-scale computer-use agent tasks with reliable rule-based evaluators is already extremely challenging and resource-intensive. Building such tasks for enterprise workflows, where domain expertise is essential, is even more difficult. Despite these constraints, we conducted comprehensive user interviews to understand how platform administrators, sales representatives, and service agents use the Salesforce platform in their day-to-day work. From these studies, we identified and refined tasks that are both representative of real enterprise workflows and meaningful for automation by computer-use agents. Through this process, we curated 300 high-quality tasks, which is comparable in scale to OSWorld (369 tasks).
>
> **We respectfully hope the reviewer will consider the task domain and difficulty when discussing task quantity**. Enterprise environments inherently involve more complex, domain-specific workflows, making task collection more demanding than in daily consumer-facing domains.
>
> We also attempted to further scale the benchmark by creating task templates and using LLMs to synthesize additional tasks and evaluators.. Unfortunately, with the most performant LLMs, most of the synthesized tasks may either lack diversity or be invaild. This highlights the difficulty of relying on automated generation for high-fidelity enterprise workflows and further underscores the significance of the curated tasks we provide.

---

> > ### Author Response · Authors · 2025-11-20
> > **Official Comment by Authors (patr 2)**
> >
> > > The task distribution is heavily skewed (57% for the 'admin'), which may limit the benchmark's ability to evaluate agent performance comprehensively across all core business roles.
> >
> > The task distribution naturally reflects the findings of our user research. One key reason for the apparent skew is that roles such as sales agents and service agents rely on a broader tool ecosystem. For example, sales agents may use HubSpot, while service agents may use Slack or other communication platforms. As a result, even if our user interviews were evenly distributed across the three roles, the set of Salesforce-specific workflows we could collect is still dominated by platform administrators, whose daily work is almost entirely performed within Salesforce.
> > In other words, the skew arises not from sampling bias but from the inherent nature of role-based workflows: administrators interact with Salesforce for most of their job functions, whereas sales and service roles use Salesforce alongside several other tools. Consequently, the collected Salesforce-grounded tasks naturally reflect this distribution.
> >
> >
> > > For the experiments, the claim “The high task success rates seen in browser-use agents can be attributed to the combination of the stronger planning ability of the foundation models, richer
> > observation space, and the more efficient action space design used in browser-use agents". This comparison between "Browser-Use" and "Computer-Use" agents seems to be methodologically unbalanced.
> >
> > Our intention was to explain the observed performance gap, not to claim that the comparison between browser-use and computer-use agents is methodologically fair. In fact, as the reviewer notes, we explicitly acknowledge in the paper that these two agent paradigms differ significantly in observation and action space design. Our discussion was meant to provide plausible factors contributing to the performance differences, rather than to assert a controlled or balanced comparison.
> >
> > > Minor: SCUBA is deeply coupled with Salesforce. While this provides depth in specific business scenarios, it sacrifices breadth. Conclusions drawn from this benchmark regarding UI nav and DOM-related stuffs may be difficult to generalize directly to other platforms.
> >
> > We do not intend to generalize our conclusions directly to other platforms. All of our findings are grounded strictly in the agents’ performance on tasks within the Salesforce environment. UI navigation on Salesforce is challenging even for humans who are unfamiliar with the platform, so it is unsurprising that agents struggle as well. At the same time, DOM-related issues are widely recognized as a fundamental engineering challenge in real-world web environments.
> >
> > At the same time, DOM-related issues are widely recognized as a fundamental engineering challenge in real-world web environments. For instance, industry discussions, such as the recent [Yutori blog post](https://yutori.com/blog/the-bitter-lesson-for-web-agents), also highlight the inherent difficulty of reliably parsing and interpreting complex DOM structures.

---

> ### Author Response · Authors · 2025-11-20
> **Official Comment by Authors (patr 3)**
>
> > The paper highlights a dramatic performance drop for agents moving from OSWorld to SCUBA, attributing it to planning / grounding capabilities. This analysis could be deeper. Does this failure simply indicate a domain shift, or does it suggest that benchmarks for general desktop applications (like OSWorld) are poor proxies for the challenges of complex, web-based, stateful enterprise applications?
>
> 1. We did not intend to attribute the performance drop from OSWorld to SCUBA solely to planning or grounding capabilities. **Rather, our comparison is meant to highlight the difficulty that current computer-use agents face in generalizing across environments.** Planning and grounding weaknesses are generic failure modes: we observe them in both OSWorld and SCUBA. This is why we state that *planning and grounding remain the primary causes of task failure*, not that they explain the cross-benchmark performance gap by themselves.
>
> 2. The difficulty in generalizing is related to how the agents are trained and the scarcity of high-quality computer-use agent trajectories (even in a general domain). The recent large-scale agent trajectory dataset `AgentNet`, where trajectories are crowdsourced, has only **27k** trajectories, which is still far from sufficient for robust generalization across environments.
>
> 3. Finally, it is important to note that SCUBA and OSWorld target different domains and evaluation goals. SCUBA focuses specifically on enterprise workflows within a complex, stateful, web-based platform, whereas OSWorld centers on general desktop applications. The performance differences, therefore, also reflect the distinct challenges inherent to these task distributions.
>
> ---
>
> We sincerely thank the reviewer for the thoughtful feedback and for engaging closely with our work. We have carefully addressed all raised concerns. We hope that the additional explanations help better convey the contributions of our work.
>
> **If the reviewer finds that these clarifications strengthen the paper and resolve earlier concerns, we would greatly appreciate a reconsideration of the overall assessment and score.**
>
> We remain very open to further suggestions that could help improve the paper. Thank you again for your time and constructive review.

---

> > ### Comment · Reviewer_YpFg · 2025-11-23
> > **Follow-up Response from Reviewer**
> >
> > Thank you for your response. I feel that the rebuttal itself has clarified some of my questions. Although I still believe the benchmark itself has shortcomings in the number of tasks, etc. (I think an enterprise workflow involves many more scenarios than the common ones faced in OSWorld).
> >
> > In the rebuttal, the authors mentioned that "The difficulty in generalizing is related to how the agents are trained and the scarcity of high-quality computer-use agent trajectories." As a dynamic environment, does Scuba itself support being used as a playground/sandbox for trajectory collection/synthesis[1] to improve the agent's in-domain performance on it? Typical open-source works like OSWorld[2], AndroidWorld[3], Webarena[4], ScienceBoard[5], etc., all support data synthesis on top of them, and I'm not sure if an enterprise work can also provide this support.
> >
> > I am also reading the questions and rebuttals raised by QXNn, X9Qy, and JVKZ. I have initially adjusted my rating from 2 to 4.
> >
> > [1] OS-Genesis: Automating GUI Agent Trajectory Construction via Reverse Task Synthesis
> >
> > [2] OSWorld: Benchmarking Multimodal Agents for Open-Ended Tasks in Real Computer Environments
> >
> > [3] AndroidWorld: A Dynamic Benchmarking Environment for Autonomous Agents
> >
> > [4] WebArena: A Realistic Web Environment for Building Autonomous Agents
> >
> > [5] ScienceBoard: Evaluating Multimodal Autonomous Agents in Realistic Scientific Workflows
> >
> > Thanks!

---

> > > ### Author Response · Authors · 2025-11-24
> > > **Follow-up Response by Authors (part 1)**
> > >
> > > We thank the reviewer for the insightful questions. Please find our response below.
> > >
> > > > Although I still believe the benchmark itself has shortcomings in the number of tasks, etc. (I think an enterprise workflow involves many more scenarios than the common ones faced in OSWorld).
> > >
> > > We do agree with the argument that enterprise workflow involves many more scenarios than the tasks in SCUBA can cover. According to our survey,  a good portion of workflows **involve multiple platforms**. Below are some concrete examples.
> > >
> > > 1. When service agents work on a case or incidents, they may use the *Salesforce platform* to track and update case details, and use their *communication tools* to communicate with the internal team for additional information confirmation,  order placement, or providing live phone support.
> > >
> > > 2. For a sales agent, it might use *HubSpot* for lead generation, and then use *LinkedIn Sales Navigator* for decision role confirmation, and finally manage lead information in the *Salesforce platform*.
> > >
> > > Even just within the *Salesforce platform*, there are additional tasks from the survey not included in SCUBA,  for example,  lead scoring, field service, account engagement, and AI-driven insights.
> > >
> > > However, **we chose not to include them in SCUBA solely because either the functionalities used in the workflows are paid features or there are no freely available sandbox environments** (and the subscription fee for some platforms can be significant).
> > >
> > > Due to the above constraints, **SCUBA made an attempt to balance the need for realistic business workflows in the real environment and the accessibility to the broader community. This trade-off limits the number of tasks we presented in this benchmark**.
> > >
> > > We are currently working on the alternatives to address workflows that involve multiple platforms. Please stay tuned.
> > >
> > >
> > > > As a dynamic environment, does Scuba itself support being used as a playground/sandbox for trajectory collection/synthesis[1] to improve the agent's in-domain performance on it?
> > >
> > > Yes, SCUBA supports trajectory collection/synthesis as it is a dynamic environment. There are actually multiple ways to achieve this.
> > >
> > > 1. Let the agent explore the environment with some general goals, like how to use search functionality and where to access the settings page. One can collect trajectories while the agent explores the environment. One caveat for this approach is that one may still craft either rule-based or LLM-driven evaluators.
> > >
> > > 2. Reuse the templates from the SCUBA to expand the task pool. Let's use a template to demonstrate `Create a {chart_type} chart named {chart_name} for the {object_name} object, under the {list_name} list view. Aggregated on the {aggregate_field} field by {aggregate_type}. The grouping field should be {grouping_field}.` Different values in the template would create different levels of GUI action complexity and diversity. We have already provided 5 instances, so one can easily collect in-domain trajectories by changing different values in this template. And the modification of the evaluators should also be straightforward since one just needs to change the ground truth value. One can even use the five instances as in-context examples to prompt the LLM to generate more tasks and evaluators. One caveat for this approach is that the synthesized values and evaluators need to be manually verified since the LLM can hallucinate. Empirically, we observe this approach works well out of the box on certain tasks, while some require significant human intervention. So it's more like leverage human-LLM-co-design to expand the task pools.
> > >
> > > 3. One can also craft new tasks that are not from the templates. This can be done by specifying the task prompt and the evaluator definition, where one can refer to the [Trailhead](https://trailhead.salesforce.com) and the Salesforce [developer doc](https://developer.salesforce.com/docs/atlas.en-us.api_meta.meta/api_meta/meta_intro.htm) for more details.

---

> > > > ### Author Response · Authors · 2025-11-24
> > > > **Follow-up Response by Authors (part 2)**
> > > >
> > > > Lastly, we would like to comment that for benchmarks like SCUBA and OSworld, adding more diverse tasks is still not trivial, since one needs to carefully consider the realism of the task,  the prerequisites of the task (some tasks need pre-populated data and data-access privilege setup), and comprehensiveness of the evaluators (so that different aspects of the taks are correctly accessed). We provide a snippet of the evaluator implementation below that corresponds to the task template mentioned in `2` to highlight the complexity of constructing a valid evaluator, where the difficulty lies in defining reasonable milestone scores, understanding the data schema in the platform, and constructing a valid Salesforce SOQL query.
> > > >
> > > > ```python
> > > > # Check if chart is created, which defines the first milestone. The query below is constructed by referring to the Salesforce Developer Documentation
> > > > query = f"SELECT Id, AggregateType, AggregateField, GroupingField, ChartType FROM ListViewChart WHERE DeveloperName='{chart_name}'"
> > > > # by running the query, we can get observed_aggregate_type
> > > > if observed_aggregate_type == aggregate_type_ground_truth:
> > > >     score += 0.2
> > > > else:
> > > >     failure_messages.append(f'Agent failed to use the correct aggregation type.')
> > > > # ... omit for simplicity ...
> > > > # Check if the chart is associated with the list view, which defines the second milestone.
> > > > query = f"SELECT Id FROM ListView WHERE Name='{list_name}'"
> > > > # ... omit for simplicity ...
> > > > ```
> > > >
> > > >
> > > > **We hope the response can clear the remaining concerns on the quality and contribution of this work, and we are happy to take more questions. We would appreciate it if the reviewer would consider updating the score in light of our response.**

---

> > > > > ### Comment · Reviewer_YpFg · 2025-11-27
> > > > > **Thanks again for your response**
> > > > >
> > > > > Thank you for your response!
> > > > >
> > > > > After reading your rebuttal, I believe that its value would be greater if we leverage scuba dynamic environment for trajectory collection (like the typical works I mentioned) than only a benchmark.
> > > > >
> > > > > I now understand the reason for the limit on the number of tasks, and I have decided to further update my rating from 4 to 6.

---

> > > > > > ### Author Response · Authors · 2025-11-29
> > > > > > **Thank you for the feedback and updated rating**
> > > > > >
> > > > > > We sincerely appreciate the reviewer’s thoughtful comments and willingness to raise their score from 2 to 6. Your constructive feedback and active engagement have greatly improved the quality of our submission.

---

### Official Review · Reviewer_X9Qy · 2025-11-07

**Soundness:** 3
**Presentation:** 3
**Contribution:** 3
**Rating:** 6
**Confidence:** 4

**Summary:**

This paper introduces SCUBA, a new benchmark designed to evaluate computer-use agents on realistic customer relationship management (CRM) workflows within the Salesforce platform. It contains 300 task instances derived from real user interviews, covering roles like administrators, sales representatives, and service agents. SCUBA features a sandbox-based Salesforce environment, rule-based milestone evaluation, and demonstration-augmented evaluation for assessing enterprise agents' performance. The experiments benchmark nine agents (browser-use and computer-use paradigms) under zero-shot and demonstration-augmented conditions, revealing substantial performance gaps between open- and closed-source models, as well as between paradigms.

**Strengths:**

- Timely and highly relevant: The paper tackles an important gap—evaluating GUI-based LLM agents on enterprise-grade workflows, rather than consumer or synthetic tasks.
- Strong realism and comprehensiveness: SCUBA operates in real Salesforce sandbox environments, using realistic personas and tasks drawn from actual user interviews.
- Clear methodological contributions: Task construction pipeline (template → paraphrase → initialization → evaluation) is well-detailed and reproducible.Fine-grained, interpretable milestone scoring system provides richer signals than binary success rates.

**Weaknesses:**

- Insufficient qualitative error analysis: While quantitative metrics are extensive, more detailed examples of failure cases (e.g., grounding errors, navigation dead ends) would better illuminate where agents struggle.
- Demonstration quality underexplored: The authors note some demonstrations are suboptimal (e.g., missing shortcuts), but the process for ensuring demonstration consistency and coverage could be better justified.

**Questions:**

How do the authors plan to maintain task validity when Salesforce updates its sandbox environment or API schema?
What percentage of failures stem from grounding vs. reasoning vs. action space limitations? Could process rewards help isolate these error modes quantitatively?
Will SCUBA support online fine-tuning or reinforcement learning in the sandbox (e.g., for ComputerRL-style training)?
Given the availability of fine-grained rewards and high-quality labeled trajectories, why did the authors not conduct fine-tuning or reinforcement learning on open-source models to build a stronger GUI agent baseline?

---

> ### Author Response · Authors · 2025-11-20
> **Official Comment by Authors (patr 1)**
>
> We appreciate the reviewer's acknowledgment of the contribution of this work, i.e., real environment, enterprise-level tasks, and fine-grained evaluators.  We would like to address the reviewer's concerns as follows.
>
> > Insufficient qualitative error analysis: While quantitative metrics are extensive, more detailed examples of failure cases (e.g., grounding errors, navigation dead ends) would better illuminate where agents struggle.
>
> We provide an expanded qualitative analysis of key failure modes by analysing different agents' trajectories. Beyond adding more illustrative examples for Observations 1–3, our broader review of trajectories also reveals three additional systemic issues (Observations 4–6) that frequently arise in long-horizon enterprise tasks.
> Please refer to the appendix of the updated paper. For the convenience of the reviewer to gain a full picture,  we summarize all qualitative failure modes below.
>
> 1. **Grounding errors remain pervasive.**
>    Agents frequently misclick or mislocate UI elements, even when their reasoning is correct, leading to systematic action failures.
> 2. **History management misleads progress tracking.**
>    Agents often assume previous actions succeeded, even when they did not. Since the failed actions are still provided as part of the history, it causes cascading errors.
> 3. **Human demonstrations are correct but not always optimal.**
>    Demonstration-based trajectories can introduce unnecessary steps or suboptimal navigation patterns that agents would follow.
> 4. **Agents fall into navigation dead ends.**
>   During the middle of the task, the agent decides to search for something; it either creates a new tab or a new browser session, hence losing the context of the task, which should happen within the Salesforce platform, and the login states. Eventually, the agent terminates with a totally wrong or irrelevant webpage.
> 5. **Lack of domain knowledge hinders enterprise navigation.**
>    Even with correct UI perception, agents fail to identify key enterprise UI elements to interact with (e.g., Salesforce Setup) for the task or sub-goal, revealing the fact that the agent does not have prior UI knowledge on the Salesforce platform.
> 6. **Agents struggle to recover from early mistakes.**
>    Once an incorrect state is created (e.g., wrong form entry), agents rarely revisit or correct it, causing unavoidable downstream failure.
>
> > Demonstration quality underexplored: The authors note some demonstrations are suboptimal (e.g., missing shortcuts), but the process for ensuring demonstration consistency and coverage could be better justified.
>
> We would like to clarify the potential misunderstanding about the `demonstration quality, consistency, and coverage`.
>
> The demonstrations are summarized from the human annotation, where annotators followed the knowledge articles to finish the tasks. The knowledge articles are of high quality and consistency, and human trajectories are also reviewed to ensure correctness. The reviewer might want to check [examples](https://trailhead.salesforce.com/content/learn/modules/lex_implementation_user_setup_mgmt/lex_implementation_user_setup_mgmt_configure_user_access-hoc) of the knowledge articles mentioned in the paper to get a better sense of the quality. For the concerns about the coverage, all the task templates have corresponding knowledge articles.
>
> For comments on "suboptimal" and "shortcuts", we mainly try to explain the observation that, in some cases, with human demonstration, the agents take more steps than the zero-setting. These shortcuts are never mentioned in the tutorial and human demonstration, yet a few agents (mainly Gemini) were able to find and exploit.

---

> > ### Author Response · Authors · 2025-11-20
> > **Official Comment by Authors (patr 2)**
> >
> > > How do the authors plan to maintain task validity when Salesforce updates its sandbox environment or API schema?
> >
> > This is a good question, and we believe the evolving nature of the live environments is a unique challenge for all live benchmarks, like Online-Mind2web[1] and WebVoyager [2].
> >
> > As we replied to the reviewer **QXNn**, we  **commit to long-term maintenance and update of this benchmark** to ensure the task validity. Apart from these human efforts, there are also some facts that make the task valid long enough.
> >
> > 1. Although Salesforce routinely updates its APIs, it offers quite robust backward compatibility support.
> > 2. When designing the tasks, we specifically chose tasks requiring features that are stable and supported in the Salesforce platform for years. For new features like Agentforce, we chose not to include them in the benchmark since UI-wise and API-wise are changing really fast.
> >
> > [1] An Illusion of Progress? Assessing the Current State of Web Agents
> > [2] WebVoyager: Building an End-to-End Web Agent with Large Multimodal Models
> >
> > > What percentage of failures stem from grounding vs. reasoning vs. action space limitations? Could process rewards help isolate these error modes quantitatively?
> >
> > We can assure that no failures stem from the action space limitations, since all the tasks are validated by the annotators, and these annotators used exactly the same action space as the agents.
> >
> > However, it is difficult to provide a concrete estimate of the proportion of failures attributable to grounding versus reasoning. In nearly all failed agent trajectories we inspected, the two types of errors are intertwined. For example, the agent may ground on the wrong element and navigate to an incorrect page, after which the model is unable to reason effectively to recover the correct environment state. Conversely, there are cases where the model fails to reason properly and thus selects the wrong element to interact with. In addition, the trajectory to a correct outcome is often not unique, making step-level diagnosis of grounding versus reasoning errors impractical. As a result, it is challenging to qualitatively separate these error modes.
> >
> > The process reward (i.e., milestone reward) also does not provide a quantitative way to isolate these errors, as it simply indicates whether a subgoal was achieved. When the agent fails to reach a subgoal, the underlying cause could be either a grounding issue, a reasoning issue, or both.
> >
> >
> > > Will SCUBA support online fine-tuning or reinforcement learning in the sandbox (e.g., for ComputerRL- style training)? Given the availability of fine-grained rewards and high-quality labeled trajectories, why did the authors not conduct fine-tuning or reinforcement learning on open-source models to build a stronger GUI agent baseline?
> >
> > Yes, like OSWorld, SCUBA supports RL training. That's also one of the reasons we use rule-based evaluators and design process reward. We indeed plan to do RL on SCUBA, but it's out of the scope of this paper. This paper aims to provide a realistic testing environment and tasks to understand where the computer-use agents are for automating the enterprise tasks.

---

### Official Review · Reviewer_QXNn · 2025-11-08

**Soundness:** 3
**Presentation:** 3
**Contribution:** 4
**Rating:** 6
**Confidence:** 3

**Summary:**

SCUBA establishes a comprehensive and enterprise-grounded framework for evaluating computer-use agents on real CRM workflows. It captures the full spectrum of enterprise challenges (UI navigation, data manipulation, workflow automation, and troubleshooting) that prior benchmarks such as OSWorld and WorkArena fail to encompass.

**Strengths:**

- The overall presentation of the benchmark—covering dataset construction, filtering, and annotation—is exceptionally coherent and methodically executed. The benchmark is clearly grounded in realistic business workflows and real-world use cases.

- The paper demonstrates clarity in explaining implementation details, e.g., the description and justification of the chosen inference engine.

- I especially appreciate that the benchmark provides rule-based evaluators with milestone (process) rewards, enabling multi-dimensional performance assessment.

**Weaknesses:**

Personally, I find this benchmark meaningful and thoughtfully described. My comments here are merely minor suggestions: (1) The authors might consider evaluating a broader range of open-source models [1,2,3]. It would also be worthwhile to update UI-TARS to the latest UI-TARS-2B version. I understand this may not have been feasible before the ICLR submission deadline, but such updates could be included in the camera-ready version.

---

[1] Aria-UI: Visual Grounding for GUI Instructions

[2] Mobile-Agent-v3: Fundamental Agents for GUI Automation

[3] MagicGUI: A Foundational Mobile GUI Agent with Scalable Data Pipeline and Reinforcement Fine-tuning

**Questions:**

- Line 261 mentions the use of ChatGPT — could the authors specify which version was employed?

 - Line 258 states that each task template is filled five times. \Were these fillings done manually or LLM-generated? Additionally, in the example where “one query might require extra scrolling or alphabet-based searching to locate the target data, while another might demand keeping track of previously deselected checkboxes,” were these variations manually created?

- I am uncertain how demonstration examples were obtained for each test sample. Were they directly taken from the two demonstrations associated with the corresponding task template? If so, how should we interpret Line 811, which states that “two samples are tested with exact demonstrations and three samples are tested with similar demonstrations”?

- Since the authors have shown that adding human demonstrations to every test sample is not always optimal, I am curious about potential gains from incorporating simple memory mechanism, similar to the agent memory module implemented in MMINA.

---

> ### Author Response · Authors · 2025-11-20
> **Official Comment by Authors (part 1)**
>
> We appreciated the reviewer's acknowledgment of the contribution of this work and the quality of the presentation. We would like to address the reviewer's concerns as follows.
>
> > The authors might consider evaluating a broader range of open-sourced models [1] Aria-UI, [2]MobileAgent, and [3]MagicGUI. It would also be worthwhile to update UI-TARS to the latest UI-TARS-2B version.
>
> We thank the reviewer for pointing out other strong open-sourced models. Due to the time constraint, we implemented and tested the following three agents: `UI-TARS-2B`, `GUI-OWL`, and `MobileAgentV3` (both are from [2]).
>
> | Method                          | Settings  | Milestone Score ↑ | Success Rate ↑ | Time (min) ↓ | Steps ↓ | Tokens (k) ↓ |
> |---------------------------------|-----------|-------------------|----------------|--------------|---------|--------------|
> | UITARS-2B                       | zero-shot | 0.02              | 0.3%           | 12.18        | 33.81   | 167.50       |
> | UITARS-2B                       | demo-aug  | 0.06              | 1.2%           | 12.09        | 34.70   | 187.21       |
> | GUI-Owl-7B                      | zero-shot | 0.05              | 1.0%           | 12.96        | 34.05   | 167.64       |
> | GUI-Owl-7B                      | demo-aug  | 0.12              | 2.7%           | 11.18        | 36.81   | 181.50       |
> | MobileAgentV3 (with GUI-Owl-7B) | zero-shot | 0.11              | 3.1%           | 21.47        | 41.96   | 755.63       |
> | MobileAgentV3 (with GUI-Owl-7B) | demo-aug  | 0.18              | 7.3%           | 19.96        | 42.98   | 830.85       |
>
> We decided not to include the results for `Aria-UI` and `MagicGUI` during the rebuttal phase for the following reasons.
>
> 1. For `Aria-UI`, it is a grounding-only model, meaning that it needs to be used in certain agentic frameworks, e.g., planner (GPT-5) + grounder (Aria-UI). We have included `Agent-S2.5` to show how the agentic system performs.  In `Agent-S2.5`, `UI-TARS-1.5` is used as the grounder.
> 2. The `MagicGUI`'s action space is for mobile, which is not the intended use cases for the Salesforce sandbox environment.
>
> We are excited to see that there's a strong growing trend in performant open-sourced CUA agents. However, due to the time and computation constraints, we cannot implement and test performant open-sourced models comprehensively. Plus, one needs to be careful on the actual implementation of the agentic loop, i.e, the configuration of each model and hyperparameters. Otherwise, the results might be skewed. Yet, some details are missing in the open-sourced codebase.
> Despite these challenges, in the rebuttal phase, we have already reached out to the teams for developing models [1] and [2] to learn more details. We plan to add [1] and similar models like Uground in the future.
>
> Finally, we would like to point out that it's on our plan to include more open-source models after the discussion phase. The sustainable plan is to reach out to teams that have developed these more and collaborate with the community to onboard these methods on this benchmark. We also commit to **long-term maintenance and update of this benchmark since it's a live environment**.
>
> > Line 261 mentions the use of ChatGPT — could the authors specify which version was employed?
>
> This description is actually not accurate. We used GPT-5's API for paraphrasing the queries. We will revise it in the camera-ready version.

---

> ### Author Response · Authors · 2025-11-20
> **Official Comment by Authors (patr 2)**
>
> > Since the authors have shown that adding human demonstrations to every test sample is not always optimal, I am curious about potential gains from incorporating simple memory mechanism, similar to the agent memory module implemented in MMINA.
>
> We would like to clarify a few things.
>
> 1. `Adding human demonstrations to every test sample is not always optimal` does not necessarily mean that, in general, human demonstration is not helpful. There can be multiple reasons leading to this observation.
>
> * The way we turn the human annotation into a demonstration can be optimized. Currently, we only use a sequence of human actions (text-modality) to obtain the demonstration. Yet, one can also include the screenshot of each action to provide image-modality context to enhance the demonstration expressiveness.
> * Human annotation is not optimal. Since annotators tend to exactly follow the steps detailed in the knowledge articles to finish the tasks, the trajectory might not be the shortest one to achieve the goal. We do observe that in the zero-shot setting, some agents can exploit the UI shortcuts to reduce the steps, while under the demonstration augmented setting, the agent follows a human trajectory, ending up with taking more steps.
> * The demonstration is of high quality, yet the agent failed to faithfully follow it. This is more frequently observed in the open-sourced model. Even when the right demonstration is provided in the context of the agents, they could fail to follow some critical steps, leading to the final failure.
>
> 2. In our agent implementation, we choose to provide all actions executed so far (text modality) and the current screenshot (image modality). This aligns with the *episodic memory* defined in **MMINA**, which states "Episodic memory temporarily holds step-by-step action trajectories, enabling the agent to recall previous actions for ongoing tasks".  Our demonstration is identical to the definition *procedural memory*  defined in  MMINA, which states "encoding the full action sequence and outcomes to refine strategies for future tasks". Interestingly, MMINA chose to include two tasks for constructing procedural memory, which is the same as our setting. And we reached the same conclusion that memory/demonstration can significantly enhance performance and adaptability.

---

> > ### Comment · Reviewer_QXNn · 2025-11-26
> >
> > I would like to thank the authors for their detailed response. I believe that such a industrial benchmark is valuable and beneficial for the community, although the underlying platform seems not as easily accessible as its competitors. I am willing to maintain my positive score.

---

### Author Response · Authors · 2025-11-29
**Note on Rebuttal Updates Following the Review Rollback**

Dear Area Chair(s),

Given this unprecedented and challenging situation, we fully understand and respect the decision made by the ICLR Program Chairs. We also sincerely appreciate the additional efforts invested in handling the submissions under the current circumstances to ensure integrity and fairness.
Regarding the rollback of reviews, we would like to summarize a few facts about our submission and the author–reviewer discussion, which we hope will assist you in reassessing our work:

1. The authors of submission 3792 have never attempted to use any leaked information in any form, and we strongly oppose such behavior. **We engaged in the author–reviewer discussion strictly in accordance with the ICLR Code of Conduct.**

2. During the discussion phase, we actively participated in conversations with all reviewers and received responses from 3 of the 5 reviewers prior to the incident. As a result, the overall score changed from `66642` to `66646` before the incident occurred. We summarize the pre-rollback discussion in the table below.


| Reviwers | Initial Score  | Main Concerns / Request                                                 | Our Response                                  | Updated Score |
|----------|----------------|-------------------------------------------------------------------------|-----------------------------------------------|---------------|
|     `QXNn` |  6             | evaluate more open-sourced agents; discuss related memory mechanism     | added more experiemnts; discuss related work  | 6             |
|     `YpFg` |  2             | 300 tasks in the benchmark is limited; benchmark infrastructure and whether support trajectory collection | explained the trade-off between realism and accessibility for constructing tasks; explained ways to collect trajectories| 6             |
|     `JVzK` |  4             | whether tasks are realistic and reasonably decomposed for agents; details of evaluators; safety concerns for rule-based evaluators |  explained the details of tasks construction and evaluators; discussed the current practice, our efforts, and future direction to improve the rule-based evaluators | 4             |

3. After the discussion, reviewer `QXNn` decided to maintain the positive score `6`.

4. After two rounds of discussion, reviewer `YpFg` raised their score from `2 → 4 → 6`.

5. After the initial round of discussion, reviewer `JVzK` acknowledged that “many of my earlier questions (e.g., about task sourcing, evaluator implementation details, and the overall setup) have been addressed clearly.” In the second round, reviewer `JVzK` further elaborated on two remaining concerns: (1) Side effects and rule-based evaluation, noting that it may not fully capture undesired agent behaviors, and (2) From human-realistic to AI-reasonable task design, expressing that some tasks may be too complex for current agents and should be further decomposed.
    * For the first concern, we clarified that tasks are constructed based on the minimal functional features provided by the platform rather than the number of UI actions required, as the former leads to more coherent and natural design.
    * For the second concern, we acknowledged that this is an under-explored yet critically important issue. In response, we (i) reviewed current benchmarking practices in evaluator design, highlighting that they share the same limitations raised by reviewer `JVzK`; (ii) outlined how our approach partially addresses this; and (iii) discussed potential extensions. We believe our detailed response effectively addressed the remaining concerns.

5. Although we did not receive responses from `X9Qy` and `Hddk` before the incident, given their initially positive evaluations, we believe our replies adequately addressed their questions and requests regarding adding qualitative error analysis, clarifying functionalities supported in SCUBA (e.g., SFT or RL), human-demonstration quality, long-term reproducibility, and providing additional details on the costs of constructing the benchmark.

Once again, we sincerely thank the ACs for taking the time to read our notes, and hope it can help you to **gain a quick picture of the entire constructive discussion phase**. **We kindly hope that you will consider these positive discussion outcomes and the consensus reached before the rollback when making your final recommendation.**


Best regards,

Authors of 3792

---

### Meta-Review · Area_Chair_p9f3 · 2026-01-04

**Summary:**

This submission presents SCUBA, a benchmark for evaluating AI agents in realistic task environments. The benchmark includes 300 tasks designed to assess agent capabilities with consideration for practical deployment scenarios.

The paper received mixed initial reviews with scores of 6, 6, 6, 4, and 2 (average: 4.8). Through the discussion phase, one reviewer raised their score from 2→4→6, resulting in scores of 6, 6, 6, 4, 6 (average: 5.6) before the review rollback incident.

**Reviewer Concerns:**

The reviewers' main concerns include:
1. Benchmark scope and scale: Initial concerns about the 300-task scale and infrastructure support for trajectory collection
2. Task realism and decomposition: Questions about whether tasks are appropriately scoped for current agent capabilities
3. Evaluation methodology: Limitations of rule-based evaluators in capturing undesired agent behaviors and side effects
4. Coverage: Requests for evaluation of additional open-source agents and discussion of memory mechanisms

The authors provided careful responses addressing:
1. Trade-offs between realism and accessibility in task construction
2, Detailed explanations of task design based on minimal functional features
3. Comparison with current benchmarking practices and their limitations
4. Additional experiments with more open-source agents
5. Discussion of related work on memory mechanisms

The authors engaged constructively with reviewers, resulting in meaningful score improvements and resolution of several concerns. While some concerns about evaluator design remain (as acknowledged to be shared challenges across the field), the authors provided thoughtful discussion of limitations and future directions.

**Reviewer Scores:**

Given the positive discussion outcomes and the consensus building toward acceptance prior to the rollback, I think the remaining 4 from jvzk would also change to a positive score, given the replies to the authors.

---

### Decision · Program_Chairs · 2026-01-26

Accept (Poster)